# UP-DOWN states and ripples differentially modulate membrane potential dynamics across DG, CA3, and CA1 in awake mice

Koichiro Kajikawa, Brad K Hulse[†], Athanassios G Siapas*, Evgueniy V Lubenov*

Division of Biology and Biological Engineering, Division of Engineering and Applied Science, Computation and Neural Systems Program, California Institute of Technology, Pasadena, United States

**Abstract** Hippocampal ripples are transient population bursts that structure cortico-hippocampal communication and play a central role in memory processing. However, the mechanisms controlling ripple initiation in behaving animals remain poorly understood. Here we combine multisite extracellular and whole-cell recordings in awake mice to contrast the brain state and ripple modulation of subthreshold dynamics across hippocampal subfields. We find that entorhinal input to the dentate gyrus (DG) exhibits UP and DOWN dynamics with ripples occurring exclusively in UP states. While elevated cortical input in UP states generates depolarization in DG and CA1, it produces persistent hyperpolarization in CA3 neurons. Furthermore, growing inhibition is evident in CA3 throughout the course of the ripple buildup, while DG and CA1 neurons exhibit depolarization transients 100 ms before and during ripples. These observations highlight the importance of CA3 inhibition for ripple generation, while pre-ripple responses indicate a long and orchestrated ripple initiation process in the awake state.

*For correspondence:
thanos@caltech.edu (AGS);
lubenov@caltech.edu (EVL)

Present address: †Janelia Research Campus, Howard Hughes Medical Institute, Ashburn, United States

Competing interest: The authors declare that no competing interests exist.

## Editor's evaluation

This paper combines intracellular and extracellular recordings in the hippocampus in awake mice to investigate the initiation of sharp wave-ripples, synchronous bursts of activity thought to support memory replay. They report a specific hyperpolarization of the pyramidal cells in the CA3 subfield while the dentate granule cells or CA1 pyramidal cells are depolarized. This paper will be of interest to system neuroscientists interested in the cellular and network mechanisms of memory formation.

## Introduction

Bidirectional interactions between the hippocampus and neocortical areas are believed to play a key role in memory consolidation (**Squire, 1992**). Hippocampal ripples are deemed essential for this process because the associated population activity reflects prior experience (**Foster, 2017**; **Kudrimoti et al., 1999**; **Lee and Wilson, 2002**; **Wilson and McNaughton, 1994**) and ripple disruption results in memory deficits (**Ego-Stengel and Wilson, 2010**; **Girardeau et al., 2009**; **Jadhav et al., 2012**). Ripples provide synchronous volleys that drive cortical targets and co-occur with distinct cortical network patterns (**Battaglia et al., 2004**; **Jiang et al., 2019**; **Ji and Wilson, 2007**; **Logothetis et al., 2012**; **Mölle et al., 2006**; **Shein-Idelson et al., 2016**; **Siapas and Wilson, 1998**; **Wierzynski et al., 2009**). In particular, ripples normally occur during slow-wave sleep and quiet wakefulness when hippocampal local field potentials (LFPs) display large-amplitude irregular activity (LIA) (**Buzsáki, 1986**;

*Jarosiewicz and Skaggs, 2004*; *Kay et al., 2016*; *O'Keefe, 1976*; *Vanderwolf, 1969*), whereas neocortical dynamics are marked by the presence of UP and DOWN states (UDS), alternating periods of elevated and depressed network activity that can also be observed under anesthesia (*Cowan and Wilson, 1994*; *Steriade et al., 1993b*; *Steriade et al., 1993c*). Neocortical and hippocampal dynamics can be coordinated via the entorhinal cortex (EC), the main gateway between neocortical areas and the hippocampus, which provides direct input to the dentate gyrus (DG), and areas CA3 and CA1 (*Amaral and Witter, 1989*; *Steward et al., 1976*; *Tamamaki and Nojyo, 1993*). Experiments in sleeping and anesthetized animals show that the EC also exhibits UDS that modulate activity across hippocampal subfields (*Hahn et al., 2012*; *Isomura et al., 2006*). However, the influence of cortical UDS on hippocampal dynamics and ripple generation in wakefulness are not well understood.

Ripples are believed to be the product of excitatory buildup in the recurrent CA3 network, culminating in a population burst that drives CA1 spiking organized by the transient ripple oscillation (*Buzsáki, 1986*; *Miles and Wong, 1983*; *Stark et al., 2014*; *Traub and Miles, 1991*). This suggests that CA3 neurons should get progressively depolarized and come closer to firing threshold through the course of the ripple buildup. Separate from the buildup itself, the processes controlling ripple initiation and termination are not fully understood. In vitro experiments indicate that ripples are initiated once stochastic fluctuations in the population firing rate of CA3 pyramidal cells exceed a threshold level (*de la Prida et al., 2006*; *Schlingloff et al., 2014*), implying that an increase in the firing of CA3 neurons should result in a corresponding increase in the rate of ripple occurrence. Recent in vitro studies have also emphasized the importance of inhibitory neurons in ripple initiation (*Bazelot et al., 2016*; *Ellender et al., 2010*; *Schlingloff et al., 2014*), while other studies suggested that area CA2 and a special class of CA3 cells play a key role in ripple initiation (*Hunt et al., 2018*; *Oliva et al., 2016*). Furthermore, there is growing evidence that the functional role of ripples as well as the mechanism of their initiation may differ across the awake and sleep states (*Middleton and McHugh, 2020*; *Oliva et al., 2016*; *Roumis and Frank, 2015*; *Tang and Jadhav, 2019*). Whole-cell recordings in awake animals have opened a window to understanding the interplay between collective network activity and membrane potential ($V_m$) dynamics of hippocampal neurons, which can reveal the nature and timing of synaptic inputs and subthreshold changes that are invisible to extracellular recordings. While recent efforts have examined the $V_m$ of CA1 neurons around ripples (*English et al., 2014*; *Hulse et al., 2016*) and the $V_m$ modulation by brain state across hippocampal subfields (*Hulse et al., 2017*; *Malezieux et al., 2020*), the subthreshold dynamics of CA3 pyramidal cells around ripples and the impact of cortical inputs on $V_m$ behavior in CA3 remain unknown.

Here we combine multisite extracellular and whole-cell recordings in awake mice to characterize and contrast the membrane potential dynamics of principal neurons in CA3 with that in the DG and CA1. Our results reveal that the membrane potential of CA3 neurons is modulated by brain state and ripples in a way that is largely opposite to the modulation of dentate granule cells and CA1 neurons. This divergent modulation across hippocampal subfields is surprising given that they are interconnected and all receive inputs from the entorhinal cortex, and offers insights into the processes of ripple initiation, build up, and termination.

## Results

We combined whole-cell recordings of principal neurons across DG (22 cells), CA3 (32 cells), and CA1 (32 cells) with multisite extracellular LFP recordings spanning the radial extent of dorsal CA1 and DG in awake head-fixed mice that were free to run on a spherical treadmill (*Figure 1*). The location, morphology, and membrane properties of the recorded cells are described in *Figure 1—figure supplements 1–4*. Below we show that entorhinal inputs to DG exhibit UP and DOWN dynamics during quiet wakefulness, with ripples occurring exclusively in the UP state. Analysis of how these brain states influence membrane potential dynamics reveals that CA3 neurons hyperpolarize in the UP state when ripples occur, in contrast to neurons in DG and CA1. We then focus on characterizing how brain state is reflected in slow $V_m$ trends around ripples. Finally, we analyze ripple triggered modulation of fast $V_m$ fluctuations which reveals a prevalence of inhibition in CA3 which grows through the ripple buildup. In contrast, DG and CA1 neurons exhibit transient depolarization not only after ripple onset but also 100 ms earlier.

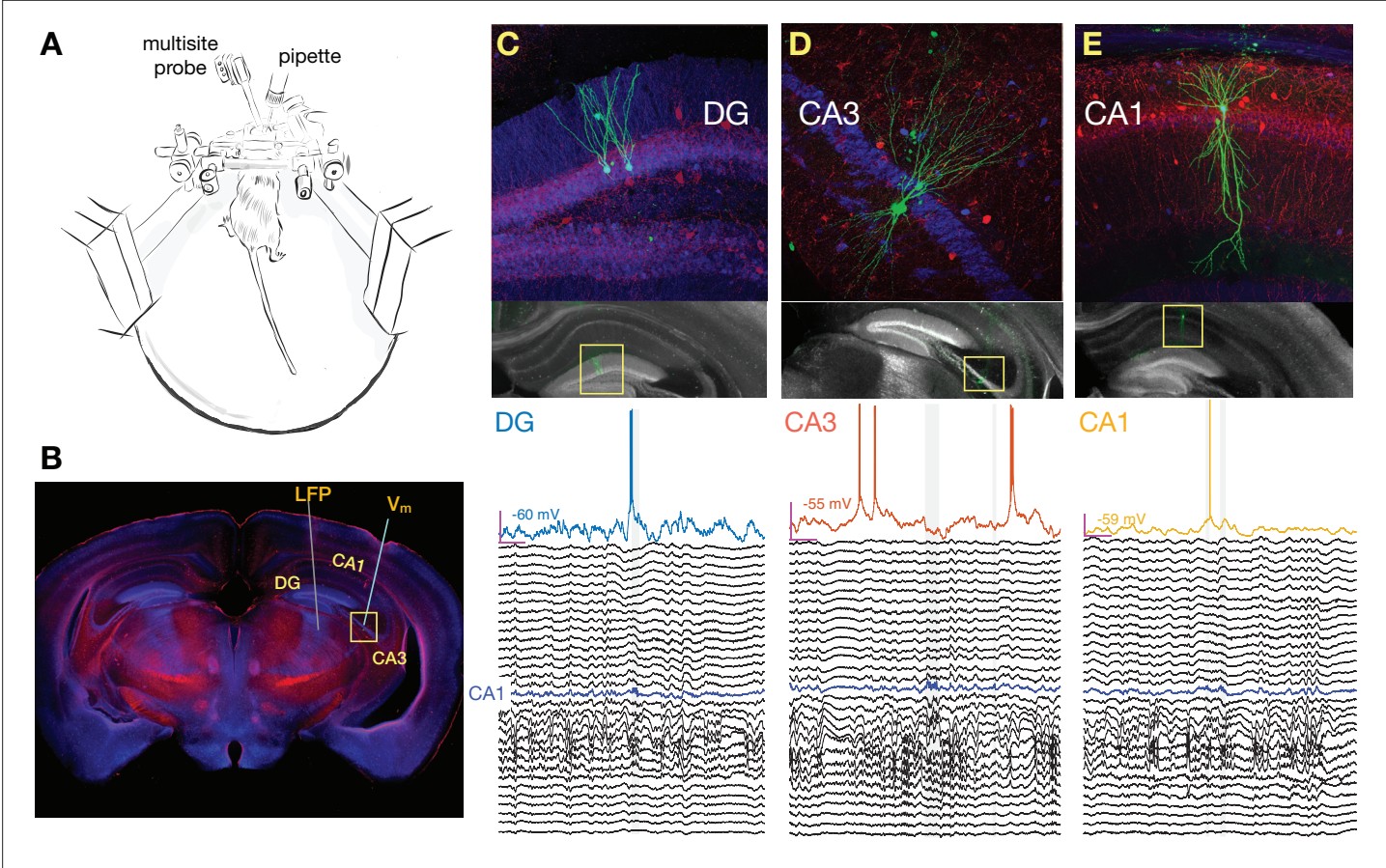

**Figure 1.** Simultaneous multisite extracellular and whole-cell recordings across hippocampal subfields. (**A**) Schematic of setup for simultaneous intracellular and extracellular recordings from awake head-fixed mice free to run on a spherical treadmill. (**B**) Typical penetration paths of multisite probe for local field potential (LFP) recordings and micropipette (targeting CA3 in this example for the neuron shown in D) for whole-cell recordings. Histological sections were stained for biocytin (green), calbindin (blue), and parvalbumin (red). (**C**) Examples of histology and recordings. Top: Recorded dentate gyrus (DG) granule cells are labeled with biocytin and their location and morphology is visualized with fluorescence microscopy. Bottom: Membrane potential (V$_m$) of a DG granule cell together with simultaneous LFP recordings spanning both CA1 and DG. The blue trace marks the pyramidal cell layer of CA1 where ripples are detected and marked by the gray vertical shading. Notice that the cell fires right before the onset of a ripple. The magenta bars indicate 200 ms and 20 mV, and the baseline V$_m$ is reported next to the trace (–60 mV). The vertical spacing between LFP traces is 1 mV. (**D**) Same as (**C**), but for a pyramidal neuron in CA3. Notice that the cell is hyperpolarized following the ripple onset. (**E**) Same as (**C**), but for a pyramidal neuron in CA1. Notice that the cell fires inside one of two nearby ripples.

The online version of this article includes the following source data and figure supplement(s) for figure 1:

**Source data 1.** Spiking and membrane properties of recorded neurons.

**Figure supplement 1.** Morphology of recorded neurons.

**Figure supplement 2.** Bursting in recorded neurons.

**Figure supplement 3.** Proximodistal locations of recorded neurons.

**Figure supplement 4.** Properties of recorded cells.

## Entorhinal inputs to DG exhibit UP and DOWN dynamics during quiet wakefulness

Based on LFP dynamics, the network state of the hippocampus can be classified as LIA, small irregular activity (SIA), or theta rhythmic, with LIA and SIA being the predominant states during quiet wakefulness (*Buzsáki, 1986*; *Jarosiewicz and Skaggs, 2004*; *Kay et al., 2016*; *O'Keefe, 1976*; *Vanderwolf, 1969*). Since this classification is typically based on a single hippocampal LFP, it does not consider the origin of the observed field fluctuations, but only their amplitude and frequency content. As a consequence, synaptic currents due to inputs from the EC as well as other hippocampal subfields

are reflected in the local field and cannot be dissociated. In order to address this, we used multisite LFP recordings and computed the laminar current source density (CSD) (*Mitzdorf, 1985*; *Pettersen et al., 2006*) throughout CA1 and DG, which in combination with the known circuit anatomy allowed us to infer the spatiotemporal pattern of synaptic activity. An example estimate of this laminar CSD is illustrated in *Figure 2*, which shows several sharp wave-ripples associated with pronounced current sinks in stratum radiatum of CA1 due to CA3 synaptic input, occurring against a background of synaptic activity in DG (see also *Figure 2—figure supplement 6*). On a longer timescale, the CSD clearly reveals alternating periods of high and low rates of transient synaptic activity lasting several seconds (*Figure 2B*) that are particularly prominent within the DG. Strikingly, these alternating periods appear to coincide with slow shifts in the membrane potential of an example CA3 pyramidal neuron (*Figure 2B*). To quantify the level of cortical input to the hippocampus, we averaged the rectified CSD over the molecular layer of DG (*Figure 2*). This measure reflects the magnitude and rate of transient synaptic currents due to inputs from layer II of the lateral and medial EC arriving at the outer two-thirds of the dentate molecular layer (*Sullivan et al., 2011*), as well as associational and return currents flowing in the inner third. Confirming our observations, the rectified DG CSD showed strong coherence with the subthreshold membrane potential of the example CA3 neuron for frequencies below 1 Hz (*Figure 2—figure supplement 1A*). This result was consistent across DG, CA3, and CA1 cells (*Figure 2—figure supplement 1B-D*).

The distribution of DG CSD activity values is fit well by a binary Gaussian mixture (*Figure 2C*) consistent with synaptic activity switching between high and low level regimes. Since EC inputs are responsible for much of the dentate synaptic currents, we identify these regimes as corresponding to entorhinal UDS. Since UDS in EC are coordinated with UDS in other cortical and thalamic areas, we reasoned that the DG CSD contains information about widespread brain state modulation and would therefore be correlated to other brain state signatures and behavioral metrics. Indeed we found that ripples tended to occur almost exclusively when DG CSD activity was high (UP state), while eyeblinks had the opposite relationship and occurred when DG CSD activity was low (DOWN state) (*Figure 2C–E*, *Figure 2—figure supplement 2*). Furthermore, the DG CSD showed a strong coherence with the slow changes (below 1 Hz) in pupil diameter (*Figure 2D*, *Figure 2—figure supplement 1A*).

In order to investigate these effects further we developed an unsupervised method for extracting UDS from the DG CSD based on a hidden Markov model (HMM) (*Figure 2C–D*). In addition to identifying state transition points, the segmentation allows the translation of the time axis into a circular UDS phase, so that event distributions can be computed with respect to the UP-DOWN phase. This analysis demonstrates that essentially all ripples occur within the UP state and the rate of ripple occurrence ramps up to a steady state value in the course of the UP state itself and abruptly terminates upon transition to a DOWN state (*Figure 2E*, *Figure 2—figure supplement 2*). It also shows that upon transition to the DOWN state the pupil quickly dilates, signaling increased arousal, and then gradually constricts through the course of the UP state, indicating a progressive reduction of arousal and attention to external stimuli through the course of the UP state (*Figure 2—figure supplement 3*). Both UP and DOWN epoch durations were distributed approximately exponentially with means of 3.06 s (UP) and 3.25 s (DOWN) and these values were consistent across recording sessions (*Figure 2—figure supplement 4*).

These observations show that quiet wakefulness can be reliably decomposed into UDS that reflect increased and decreased EC inputs, respectively. This UDS classification reflects the dynamics of the cortical input to the hippocampus, in contrast to LIA/SIA segmentation that is influenced by the states of the hippocampal subfields themselves, which as we describe below are not necessarily coherent. Nevertheless, LIA→SIA transitions mapped closely to UP→DOWN transitions, with SIA→LIA transitions also concentrated around DOWN→UP transitions (*Figure 2—figure supplement 8*). These observations are consistent with UP states broadly overlapping with LIA and DOWN states with SIA.

## CA3 membrane potential shifts are negatively correlated with entorhinal inputs to DG

What is the impact of entorhinal inputs on the activity of hippocampal neurons? The subthreshold membrane potential of the example CA3 neuron in *Figure 2B* is clearly related to the DG CSD and consequently to UP-DOWN states. Importantly, the relation is evident in the slow shifts in $V_m$ and not

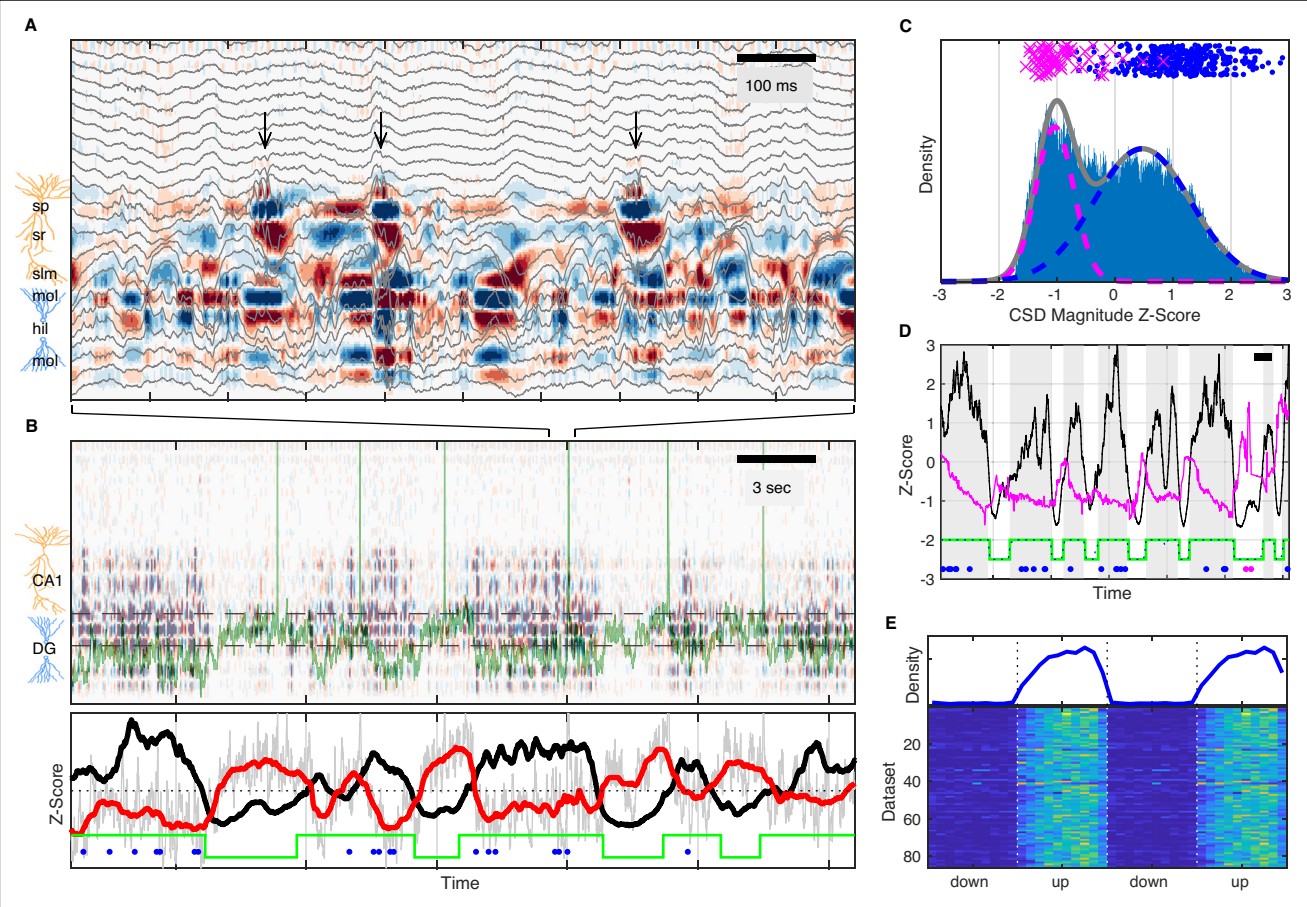

**Figure 2.** UP and DOWN states modulate slow $V_m$ shifts and ripple occurrence. (**A**) Image of current source density (CSD) derived from the local field potential (LFP) traces (gray). Ripples, high-frequency oscillations indicated by the black arrows, are associated with current sinks (red) in stratum radiatum (sr) below the pyramidal cell layer (stratum pyramidale, sp). The bottom third of the image shows large current sources (blue) and sinks (red) within the molecular layers (mol) of dentate gyrus (DG). (**B**) Top: Image of the CSD on a longer timescale reveals alternating periods of high and low CSD activity. The two interrupted black lines mark the vertical extent of the suprapyramidal molecular layer of DG. The $V_m$ of a CA3 neuron is superimposed in green. Notice that periods of low DG CSD activity (light colors) are associated with $V_m$ depolarization. Bottom: DG CSD activity (black), quantified by averaging the rectified CSD over the molecular layer of DG and smoothing, normalized to a z-score. Subthreshold $V_m$ (gray) for the CA3 neuron and its slow component (red) plotted as z-scores. Notice that the black and red traces are anti-correlated. The green stairstep trace marks epochs of elevated DG CSD activity (UP states) and decreased DG CSD activity (DOWN states). Ripples (blue dots) occur in the UP state. (**C**) Distribution of DG CSD activity fitted with a two component gaussian mixture. DG CSD activity at ripples (blue dots) and eye blinks (magenta), show preferential association with the UP and DOWN components, respectively. (**D**) Hidden Markov model (HMM) state detection based on DG CSD activity (black). DG CSD activity is high in the UP state (gray stripes), while the pupil (diameter in magenta) dilates at the onset of the DOWN state and then gradually constricts in the course of the UP state. Ripples and eye blinks are marked by blue and magenta dots, respectively. Horizontal scale bar is 3 s long. (**E**) (Top) Population average probability density of ripple occurrence as a function of UP and DOWN states (UDS) phase. Notice that ripples occur almost exclusively in the UP state. (Bottom) Rows in the pseudocolor image show the density of ripple occurrence for each dataset (n=86). Densities are replotted over two UDS cycles.

The online version of this article includes the following source data and figure supplement(s) for figure 2:

**Source data 1.** Probability density of ripple occurrence as a function of UDS phase for each recording.

**Figure supplement 1.** Coherence between rectified dentate gyrus (DG) current source density (CSD) and $V_m$.

**Figure supplement 2.** UP and DOWN state (UDS) modulation of ripple and blink occurrence.

**Figure supplement 3.** Pupil diameter around UP and DOWN transitions.

**Figure supplement 4.** UP and DOWN epoch durations.

**Figure supplement 5.** Decomposing membrane potential traces into slow and fast components.

**Figure supplement 6.** Ripple-triggered average current source density (CSD).

**Figure supplement 7.** Rectified current source density (CSD) triggered on UP/DOWN transitions and ripples.

**Figure supplement 8.** Comparison of large-amplitude irregular activity/small irregular activity (LIA/SIA) and UP/DOWN transitions.

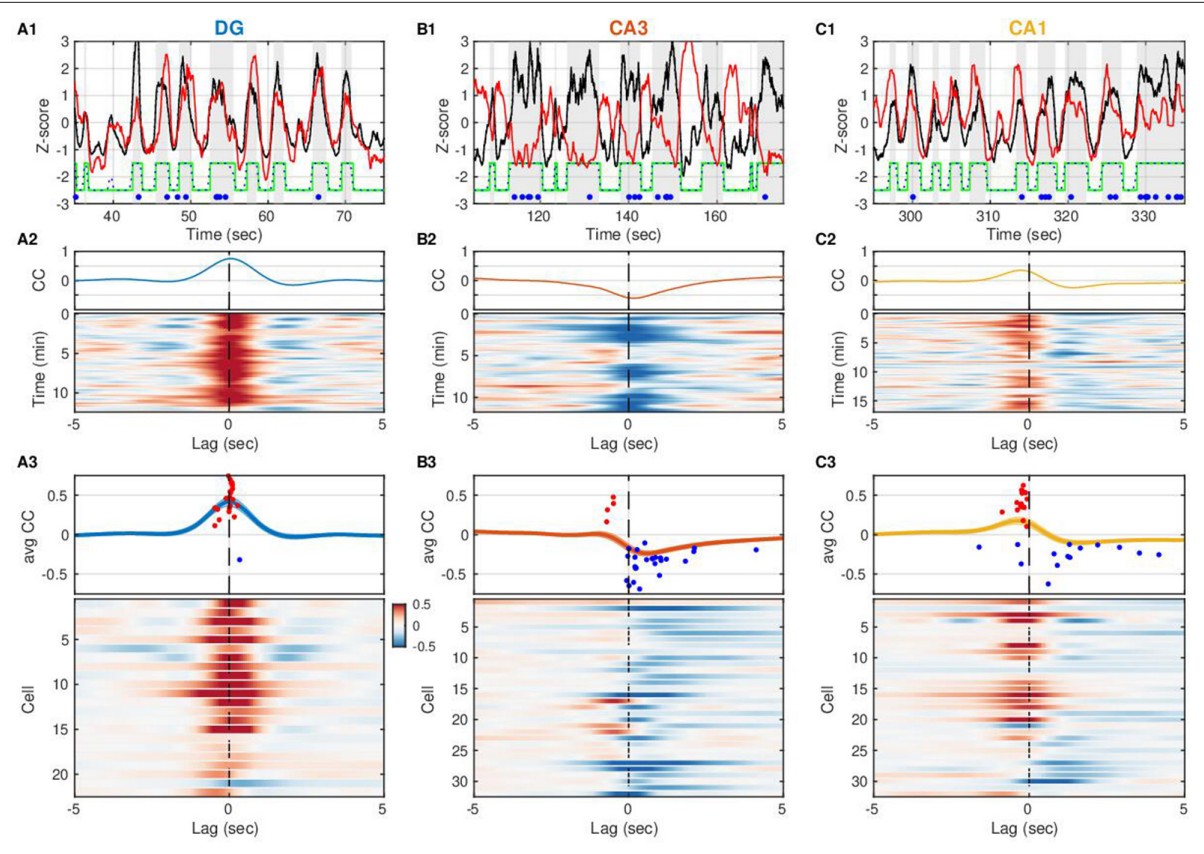

**Figure 3.** Entorhinal input to dentate gyrus (DG) is negatively correlated with slow $V_m$ shifts in CA3 in contrast to DG and CA1. (**A1**) DG current source density (CSD) activity (black) and slow component (<1 Hz) of subthreshold $V_m$ (red) for an example DG granule cell. Gray vertical stripes and green staircase trace mark periods classified as UP states. Ripples are marked by the blue dots. Notice that the $V_m$ slow component is modulated in lockstep with the DG CSD. (**A2**) (Top) Cross-covariance between the slow $V_m$ component of the example cell above and DG CSD activity. (Bottom) Cross-covariances are computed over 30 s sliding windows and displayed as a pseudocolor image. (**A3**) (Top) Population average cross-covariance of all recorded DG granule cells. Bands around the mean curves show the standard error of the mean (SEM). Dots mark the peak (red) or trough (blue) lag and amplitude of individual cells' cross-covariance extrema. (Bottom) Cross-covariances for all DG granule cells stacked vertically and displayed as a pseudocolor image. Notice that most traces are peaked near zero lag. In all figures cells are ordered by their ripple-triggered average response (RTA) rank (*Figure 6*), unless stated otherwise. (**B**) Same as (A), but for CA3 pyramidal neurons. Notice that the example cell in (**B1, B2**) and the population overall (**B3**) have membrane potentials that are anti-correlated with DG CSD activity. (**C**) Same as (A) and (B), but for CA1 pyramidal neurons. The $V_m$ of many CA1 neurons is positively correlated with DG CSD activity, but the $V_m$ (red) leads the DG CSD (black) as in (C1), so correlation peaks occur at negative lags (**C3**). A subset of CA1 neurons exhibits negative correlations at positive lags.

The online version of this article includes the following source data and figure supplement(s) for figure 3:

**Source data 1.** Cross correlation between the slow $V_m$ component and dentate CSD activity for each neuron.

**Figure supplement 1.** Strength and direction of the correlation between $V_m$ slow component and dentate gyrus (DG) current source density (CSD) activity.

---

in the superimposed faster fluctuations. This is confirmed by the fact that all significant coherence between $V_m$ and rectified DG CSD occurs below 1 Hz (*Figure 2—figure supplement 1*). We therefore separated the fast from the slow dynamics of the subthreshold membrane potential ($V_m$) fluctuations with a cutoff frequency of approximately 1 Hz (*Figure 2—figure supplement 5*).

Surprisingly, the membrane potential of the CA3 pyramidal neuron in *Figure 2B* is more hyperpolarized when DG CSD activity and hence the excitatory EC input rate is high, and conversely, more depolarized when EC input rate is low. Does EC input impact subthreshold activity in other hippocampal subfields in a similar way? To address this, we computed the cross-covariance between DG CSD activity and the slow $V_m$ component of cells in DG, CA3, and CA1 (*Figure 3*). The majority of DG cells (21/22) displayed a positive correlation to EC inputs, the majority of CA3 neurons (26/32) exhibited a negative correlation, while CA1 neurons were split in half. In other words, while the slow $V_m$

fluctuations in DG are nearly in sync with the DG CSD activity (*Figure 3A and C*), they are anticorrelated in CA3 (*Figure 3B*). Furthermore, a closer look at the lag associated with peak absolute correlation reveals that while DG neurons follow DG CSD activity very closely (37 ms median lag), the significantly correlated CA1 neurons actually lead DG CSD activity (–225ms median lag), while the trough of the CA3 negative correlation follows DG CSD activity (296 ms median lag) (*Figure 3—figure supplement 1*). This subfield ordering is inconsistent with a simple feedforward activation along the trisynaptic circuit in the awake state.

How well can we estimate the slow $V_m$ component of hippocampal neurons from the DG CSD? All hippocampal subfields receive direct input from the entorhinal cortex (EC), and yet the pattern of modulation by the UP-DOWN state cycle varies across subfields. In order to understand how EC inputs influence hippocampal neurons we used the measured DG CSD as a proxy of EC input and estimated linear transfer models for each cell treating DG CSD activity as input and the cell's slow $V_m$ component as the model output (*Figure 4*). We considered a class of finite impulse response (FIR) models allowing for non-zero filter values at negative lags and hence for a non-causal influence of the input on the output. The estimated impulse responses (*Figure 4A2–D2*) revealed that while DG granule cells had causal responses, many CA3 and CA1 cells showed positive filter values at negative lags (up to –250 ms) signaling a non-causal relation between DG CSD and $V_m$. This was consistent with the positive/negative peak correlation lags observed for DG/CA1 cells (*Figure 3—figure supplement 1A-B*), but revealed a non-causal effect in CA3 which was not evident in the cross-covariance analysis. At positive lags the impulse responses for the majority of DG and CA1 cells were positive, while they were negative (inhibitory) for the majority of CA3 pyramidal neurons. The estimated models allowed us to simulate the step responses of hippocampal neurons (*Figure 4A1–D1*) and showed that as a population DG and CA3 exhibited persistent depolarization and hyperpolarization in response to sustained EC input, respectively, while CA1 showed a transient depolarizing response that decayed within 1 s. We also simulated the slow $V_m$ components of hippocampal neurons from DG CSD and found good qualitative agreement with the experimental observations (*Figure 4—figure supplement 1*), confirming that the

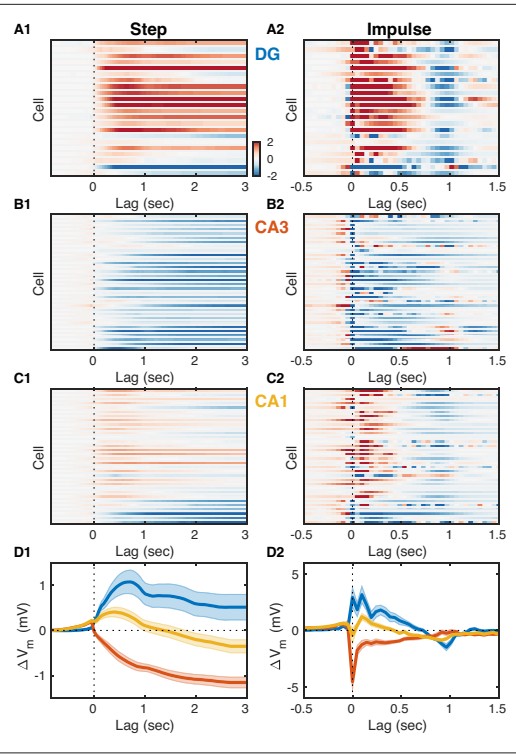

**Figure 4.** Entorhinal input differentially modulates slow $V_m$ shifts across hippocampal subfields. (**A1**) Each row of the pseudocolor image shows the step response of a linear transfer model describing the effect of dentate gyrus (DG) current source density (CSD) activity on the slow $V_m$ component for a given DG cell. The vertical interrupted line marks the onset of the input step. (**A2**) Each row shows the impulse responses of the corresponding models in (A1). The vertical interrupted line marks the onset of the impulse. Notice that the majority of DG cells exhibit causal behavior, i.e., the impulse response is near zero for negative lags. (**B**) Same as (A), but for CA3 pyramidal neurons. Notice that the majority of CA3 neurons hyperpolarize in response to entorhinal cortex (EC) input and some cells exhibit non-causal impulse responses, i.e., some impulse responses have non-zero (positive) values at negative lags. (**C**) Same as (A) and (B), but for CA1 pyramidal neurons. Notice that some CA1 cells also exhibit non-causal impulse responses. (**D**) Area-specific population average step response (**D1**) and impulse response (**D2**) color-coded by brain area. Bands around the mean curves show the SEM. Notice the distinct responses across hippocampal subfields.

The online version of this article includes the following source data and figure supplement(s) for figure 4:

**Source data 1.** $V_m$ step and impulse response to dentate CSD activity for each neuron.

**Figure supplement 1.** Linear prediction of slow $V_m$ component from dentate gyrus (DG) current source density (CSD) activity.

transfer models provide a succinct description of the behavior of hippocampal neurons in response to changing EC input levels.

## CA3 neurons hyperpolarize during UP states

The slow $V_m$ component and firing rate of hippocampal neurons were modulated at UP and DOWN transitions (*Figure 5—figure supplement 1*) in a way that was consistent with the transfer model predictions (*Figure 5—figure supplement 2*). To understand their behavior through the course of the UDS cycle, we analyzed the subthreshold membrane potential and spiking of hippocampal neurons as a function of the UDS phase (*Figure 5*). In particular, we computed the distribution of $V_m$ values conditioned on the phase of the UDS cycle. These distributions displayed phase-dependent $V_m$ means for the majority (82/86) of hippocampal neurons (*Figure 5A1–D1*, *Figure 5—figure supplement 5*) and many cells (42/86) also exhibited phase-dependent fast $V_m$ component variance (*Figure 5A2–D2*, *Figure 5—figure supplement 5*). Several important differences between the hippocampal subfields were evident. Dentate granule cells were quite homogeneous in their behavior through the course of the UP-DOWN state and all but one cell showed sustained $V_m$ depolarization which was maintained through the course of the UP state and was mirrored by a persistent hyperpolarization throughout the DOWN phase (*Figure 5A1*, *Figure 5—figure supplement 5A*). In contrast, about a third (9/32) of CA3 neurons exhibited the exact opposite behavior: sustained $V_m$ hyperpolarization through the UP state and depolarization through the DOWN state (*Figure 5B1*, *Figure 5—figure supplement 5A*). The remaining population exhibited a more transient depolarization with a peak slightly preceding or coincident with the UP transition point (*Figure 5B1*). The CA1 pyramidal cell population displayed a level of diversity that was intermediate to that of DG and CA3. About 40% of CA1 neurons (13/32) were depolarized in the UP state and hyperpolarized in the DOWN state, but the responses were more transient than in DG, with depolarization/hyperpolarization decaying through the course of the UP/DOWN state, respectively (*Figure 5C1*). The behavior of the remaining CA1 neurons appeared similar to that of the CA3 population. With respect to $V_m$ fluctuations, DG granule cells exhibited the highest variability, exceeding that in CA3 and CA1 (*Figure 5—figure supplement 3A1-D1*). With respect to overall $V_m$ fluctuations only CA1 neurons displayed state-dependent modulation of $V_m$ variability (*Figure 5—figure supplement 3D1*), with the UP state being associated with more variable $V_m$. Focusing on the fast $V_m$ component alone, both CA1 neurons (14/32) and DG granule cells (9/22) showed a significant jump in variability during the UP state with no granule cells and only three CA1 neurons having the opposite trend (*Figure 5A2–D2*, *Figure 5—figure supplement 5C*). In contrast, CA3 pyramidal neurons exhibited both jumps and drops and at the population level had a nearly constant level of $V_m$ variability throughout the UP-DOWN state cycle, which equaled that of CA1 in the UP state, but exceeded it in the DOWN phase (*Figure 5B2 and D2*, *Figure 5—figure supplement 3D1*). Interestingly, with respect to spiking two thirds of the population of DG (15/22) and CA1 (22/32) neurons exhibited similar behavior, a significant increase in firing rate that persisted through the course of the UP phase (*Figure 5A3–D3*, *Figure 5—figure supplement 5D*). In contrast, the firing behavior of CA3 neurons largely mirrored the behavior of the $V_m$ mean, with about half of CA3 neurons (17/32) exhibiting a significant increase in firing rate in the DOWN state and the remaining population showing a transient increase peaking near the UP transition point (*Figure 5B3 and D3*). Thus CA3 neurons were maximally hyperpolarized and had lowest firing probability at the end of the UP phase when the rate of ripple occurrence was highest (*Figure 2E*).

## Inhibition dominates CA3 subthreshold behavior near awake ripples

It has long been appreciated that awake ripples preferentially occur during periods of quiet wakefulness and LIA. Furthermore, here we showed that when quiet wakefulness is segmented into periods of UDS, essentially all ripples occur in the EC UP state (*Figure 2E*). The prevailing view is that ripples are the result of self-organized population bursts of activity that build within the recurrent CA3 network and then are transmitted to and further patterned by the CA1 network, where the ripple oscillation itself is readily observable in the local field near the pyramidal cell layer (*Figure 2A*; *Buzsáki, 1986*; *Miles and Wong, 1983*; *Stark et al., 2014*; *Traub and Miles, 1991*). How do neurons across the different hippocampal subfields participate in this process? To address this question we detected ripple onset times from the CA1 local field near the pyramidal cell layer and determined the ripple-triggered average (RTA) $V_m$ and firing rate modulation for each recorded cell (*Figure 6*,

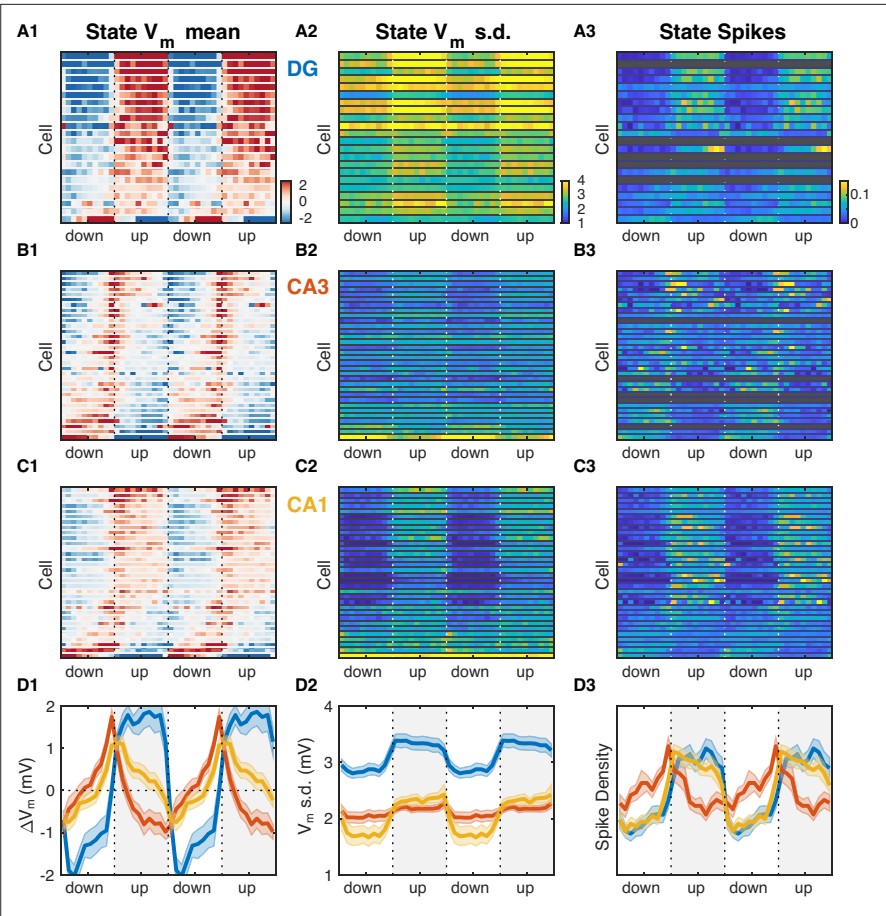

**Figure 5.** Hippocampal subfields exhibit distinct activity profiles over the UP and DOWN states (UDS) cycle. (**A1**) $V_m$ mean as a function of UDS phase for each dentate gyrus (DG) granule cell displayed as a row in the pseudocolor image. Cells are ordered by the first principal component coefficient of the image matrix (UDS rank). (**A2**) Fast $V_m$ component standard deviation as a function of UDS phase for DG granule cells. (**A3**) Observed spiking probability for all DG granule cells displayed as a pseudocolor image. Grayed out rows correspond to cells that fired fewer than 100 spikes. (**B**) Same as (A), but for CA3 pyramidal neurons. Notice that CA3 neurons are maximally hyperpolarized and have lowest firing probability at the end of the UP phase when the rate of ripple occurrence is highest (**Figure 2E**). (**C**) Same as (A) and (B), but for CA1 pyramidal neurons. (**D**) Area-specific population averages color-coded by brain area. (**D1**) Population average $V_m$ means. (**D2**) Population average fast $V_m$ component standard deviations. (**D3**) Population average spiking probability. Bands around the mean curves show the SEM.

The online version of this article includes the following source data and figure supplement(s) for figure 5:

**Source data 1.** $V_m$ and spiking responses across UP and DOWN states for each neuron.

**Figure supplement 1.** $V_m$ and spiking responses to UP and DOWN states (UDS) transitions reveal ordering in subfield activation.

**Figure supplement 1—source data 1.** $V_m$ and spiking responses to UDS transitions for each neuron.

**Figure supplement 2.** $V_m$ and transfer model responses at UP and DOWN state (UDS) transitions.

**Figure supplement 2—source data 1.** Slow $V_m$ and transfer model responses to UDS transitions for each neuron.

**Figure supplement 3.** UP and DOWN state (UDS) modulation of $V_m$ fluctuations.

**Figure supplement 3—source data 1.** $V_m$ fluctuations as a function of UDS phase for each neuron.

**Figure supplement 4.** Resting potential and proximodistal location as predictors of UP and DOWN state (UDS) modulation.

**Figure supplement 5.** Statistical differences in $V_m$ mean, fast $V_m$ variance, and firing rate across UP and DOWN states (UDS).

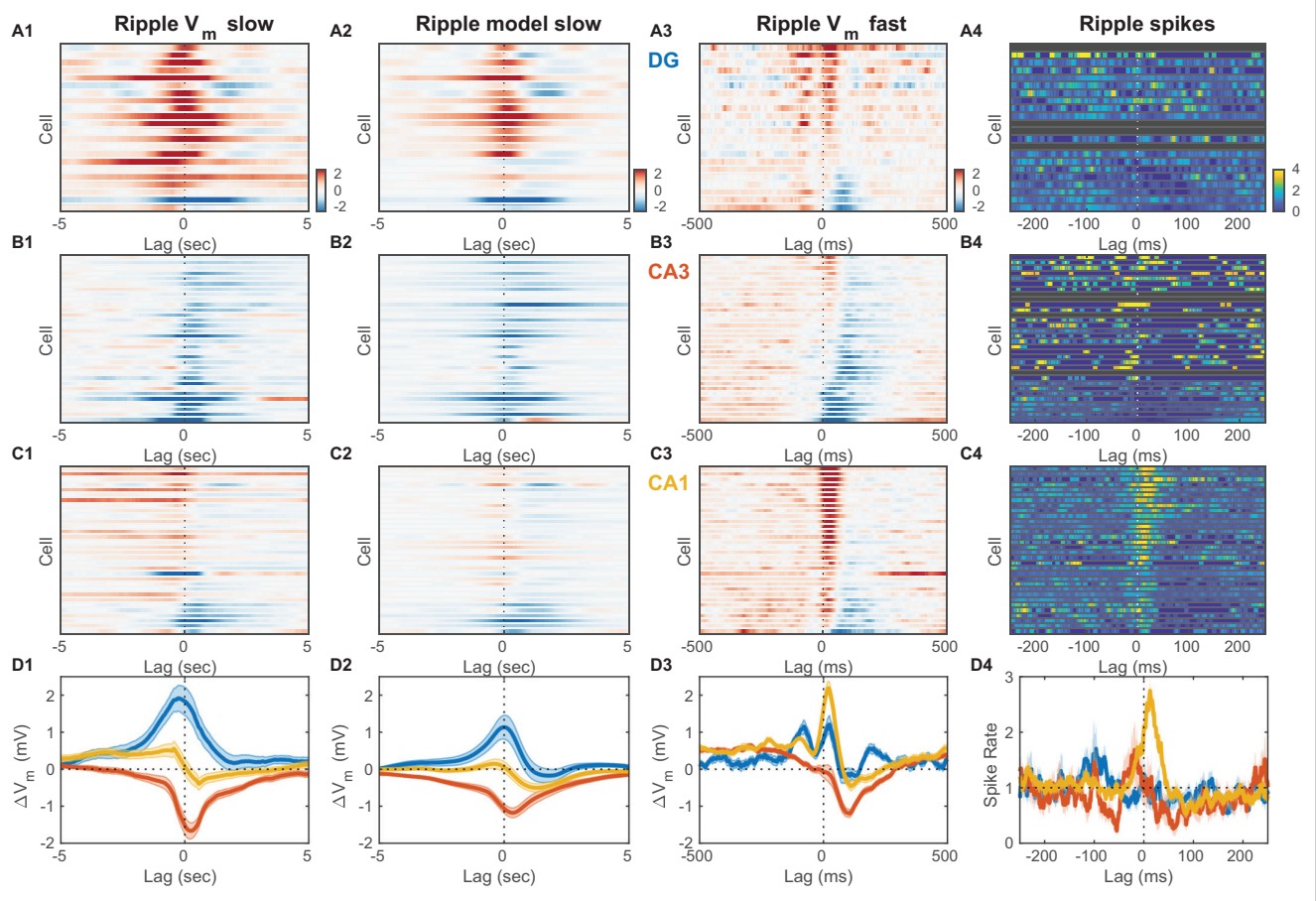

**Figure 6.** Inhibition marks slow and fast $V_m$ responses near ripples in CA3, unlike dentate gyrus (DG) or CA1. (**A1**) Mean slow $V_m$ component triggered on ripple onset for each DG granule cell displayed as a row in the pseudocolor image. Interrupted vertical line marks ripple onset. (**A2**) Transfer model predicted ripple-triggered slow $V_m$ component as in (A1). (**A3**) Mean fast $V_m$ components triggered on ripple onset and displayed as in (A1). Notice the two depolarizing peaks at −100 ms and right after ripple onset. Cells in panels (A1–3) are ordered by the first principal component coefficient of the (A3) image matrix (ripple-triggered average [RTA] rank). (**A4**) Spiking response to ripple onset for DG cells normalized by baseline firing rate. (**B**) Same as (A), but for CA3 pyramidal neurons. Notice the prominent hyperpolarization for most CA3 neurons present in both the slow and fast $V_m$ component responses. (**C**) Same as (A) and (B), but for CA1 pyramidal neurons. Notice the prominent depolarization for most CA1 neurons. (**D**) Area-specific population average slow $V_m$ response (**D1**), transfer model predicted slow $V_m$ responses (**D2**), fast $V_m$ response (**D3**), and spiking response (**D4**) color-coded by brain area. Bands around the mean curves show the SEM. Notice that modulation by UP and DOWN states (UDS) accounts for the shape of the ripple-triggered $V_m$ response on the timescale of seconds (compare [D1] and [D2]).

The online version of this article includes the following source data and figure supplement(s) for figure 6:

**Source data 1.** $V_m$ and spiking responses triggered on ripple onset for each neuron.

**Figure supplement 1.** Resting potential and proximodistal location as predictors of ripple modulation.

**Figure supplement 2.** Ripple modulation around ripple peak power.

**Figure supplement 2—source data 1.** $V_m$ and spiking responses triggered on ripple peak power for each neuron.

**Figure supplement 3.** Intrinsic properties and UP and DOWN state (UDS) modulation of CA3 cells that depolarize during ripples.

*Figure 6—figure supplement 2*). Since the RTA of the $V_m$ is the sum of the RTAs of the fast and slow $V_m$ components, we determined these average responses separately for each cell and component (*Figure 6A1–D1 and A3–D3*). The RTAs of the slow $V_m$ component showed pronounced depolarization in DG with a peak before the ripple onset, a hyperpolarization in CA3 with a trough following the ripple onset, and mixed responses in CA1 (*Figure 6A1–D1*). Since ripples occurred exclusively during the UP state and the slow $V_m$ component of hippocampal neurons was strongly influenced by state, we hypothesized that the RTA slow $V_m$ response depended strongly on the pattern of EC input and

how each cell was impacted by it. We therefore simulated the slow $V_m$ component for each cell using the linear transfer models we had estimated (*Figure 4*) and computed the RTA of the simulated slow $V_m$ response (*Figure 6A2–D2*). The RTAs of the observed and simulated slow $V_m$ components showed good qualitative agreement indicating that on the timescale of seconds $V_m$ trends near ripples are strongly influenced by the pattern of EC UDS transitions.

In contrast, the mean of the fast $V_m$ component is very weakly influenced by state and therefore its RTA should reflect synaptic activity consistently timed with respect to the ripple onset. Consistent with previous studies, we found that the fast RTA $V_m$ waveform for most CA1 pyramidal neurons had a sharp prominent peak following the ripple onset, followed by a steep return to baseline or hyperpolarization within a 100ms (*Figure 6C3*; *Hulse et al., 2016*). This behavior was mirrored in the firing rate modulation of CA1 cells (*Figure 6D3*) and is consistent with the notion that CA1 exhibits a highly synchronous population burst associated with ripple oscillations.

Since the population event in CA1 is presumably due to a self-organized population burst in CA3, one might expect that the fast RTA waveforms in CA3 should resemble those in CA1, but this was not the case (*Figure 6B3*). While a minority CA3 neurons (9/32) exhibited a small depolarizing peak near the ripple-onset, the most consistent feature of the fast RTA $V_m$ waveform in CA3 was the prominent hyperpolarization that reached a trough roughly 100 ms following ripple onset and recovered within 300 ms. The population average response shows that the decrease in $V_m$ had begun as early as 150 ms prior to the ripple onset (*Figure 6D3*). Similarly, while several CA3 neurons showed an increase in firing rate just prior to the ripple onset, about a third of the CA3 population exhibited a reduction in firing rate before and especially following the ripple onset (*Figure 6B4 and D4*). These data demonstrate that extensive inhibition, rather than excitation, is the hallmark of CA3 subthreshold activity before and after ripple bursts. It is worth noting that most neurons, even with hyperpolarizing average response, experienced depolarization and firing during a subset of ripples, consistent with a sparse activation of CA3 during ripples. Further analysis of the CA3 cells with a depolarizing peak near the ripple onset (9/32 cells) revealed that they do not have significantly different resting $V_m$, spike threshold, burst index or spikes per burst compared to the rest of the CA3 population (*Figure 6—figure supplement 3A, B*). However, these CA3 cells had a lower firing probability in the DOWN state compared to the rest of the CA3 population.

Finally, while the dentate gyrus has not been traditionally considered to be instrumental to the process of ripple generation, the fast RTA $V_m$ waveforms of granule cells showed clear modulation with about half of the population exhibiting a depolarization peak following ripple onset with a similar time course to that seen in CA1 (*Figure 6A3*). Interestingly, about half of DG granule cells also showed another depolarization peak occurring 100 ms prior to ripple onset (*Figure 6A3 and D3*). Upon closer examination a similarly timed depolarizing peak could be seen in the CA1 response prior to ripple onset, although of much smaller magnitude in comparison to the post-ripple depolarization. Similarly, a sharp wave of smaller amplitude is also consistently observed 100ms before ripple onset (*Hulse et al., 2016*). The two peaks were also present in the rectified DG CSD triggered on ripple onset (*Figure 2—figure supplement 7A3*). Since ripples can occur in close succession at intervals close to 100ms, we wondered if such ripple bursts could contribute to the observed pre-ripple depolarization. To test this hypothesis we compared the fast RTA $V_m$ aligned to the onset of isolated single ripples with that aligned to the onset of ripple doublets. The pre-ripple depolarization in DG and CA1 was similar for isolated ripples and ripple doublets arguing against the hypothesis that pre-ripple responses were due to ripple bursts (*Figure 7—figure supplement 2*). These data indicate that DG should not be viewed as a passive bystander in the awake ripple generation process and that the ripple proper is preceded by coordinated activity not only in CA3, but also in CA1 and DG.

What circuit mechanisms may be responsible for the observed subthreshold activity around ripples? Since CA3 is considered to play a key role in the process of ripple generation, we reasoned that influences on subthreshold activity originating in CA3 should scale in proportion to the CA3 population burst size. We inferred this size indirectly from the amplitude of the ripple-associated sharp wave in stratum radiatum of CA1 or equivalently by the magnitude of the underlying synaptic current due to Schaffer collateral activation (*Mizunuma et al., 2014*). We then divided the ripples from each recording session in two halves, big and small, depending on whether they were associated with above-median or below-median CA3 burst size. Finally, we computed the fast RTA $V_m$ waveform separately for big and small CA3 events and compared the results (*Figure 7*). The size of the CA3

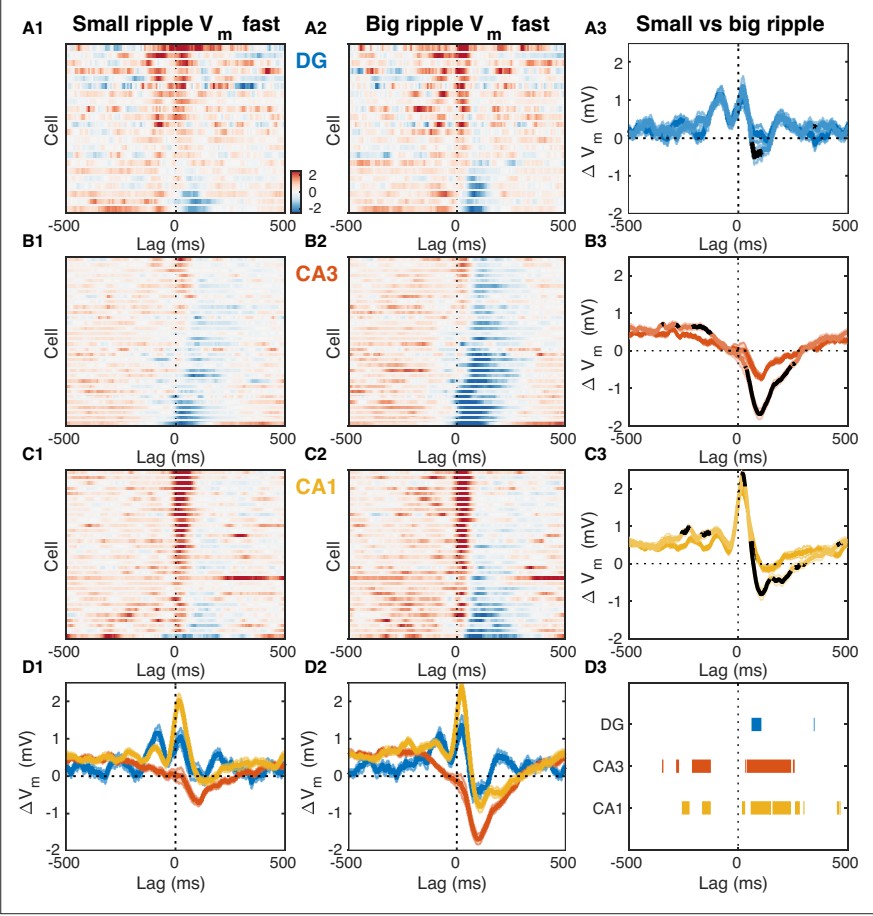

**Figure 7.** Fast $V_m$ inhibitory responses to ripples scale with CA3 population burst size. (**A1**) Mean fast $V_m$ components of dentate gyrus (DG) granule cells triggered on ripples with small (below average) sharp-wave (SPW) amplitudes. (**A2**) Same as (A1), but for ripples with big (above average) SPW amplitudes. (**A3**) Comparison of the DG population average response to small and big SPW ripples. Bands around the mean curves show the SEM. Responses to big ripples are shown in darker blue and overlap the responses to small ripples for most time lags. For significant differences the responses to big ripples are highlighted in black (paired t-test, p<0.01). (**B**) Same as (A), but for CA3 pyramidal neurons. (**C**) Same as (A) and (B), but for CA1 pyramidal neurons. (**D**) Area-specific population average responses to small (**D1**) and big (**D2**) SPW ripples. Bands around the mean curves show the SEM. (**D3**) Horizontal bars mark the onset and duration of significant differences between responses to small and big SPW ripples within each area (paired t-test, p<0.01).

The online version of this article includes the following source data and figure supplement(s) for figure 7:

**Source data 1.** Fast $V_m$ components triggered on onset of ripples with small and large amplitudes.

**Figure supplement 1.** Fast $V_m$ inhibitory responses to ripples scale with CA3 population burst size.

**Figure supplement 1—source data 1.** Fast $V_m$ components triggered on peak ripple power of ripples with small and large amplitudes.

**Figure supplement 2.** Subthreshold ripple modulation around isolated ripples and ripple doublets.

**Figure supplement 2—source data 1.** Fast $V_m$ components triggered on the onset of isolated ripples and ripple doublets for each neuron.

burst had little effect on the DG RTA waveform, apart from a slight increase in hyperpolarization about 100ms following ripple onset that was associated with the bigger events (*Figure 7A1–A3*). In contrast, subthreshold activity in CA3 clearly reflected the size of the CA3 population event (*Figure 7B1–B3*). Larger CA3 events were associated with more sustained depolarization up to 300ms prior to the ripple onset and deeper hyperpolarization following (*Figure 7B3*). In CA1, bigger CA3 events were associated with slightly more depolarization up to 250ms prior to the ripple onset and during the post-ripple

peak, but the most notable difference was the increased hyperpolarization about 100ms following the ripple onset (*Figure 7C1–C3*). These data indicate that feedback inhibition within CA3 is most likely responsible for the subthreshold hyperpolarization seen in the fast $V_m$ component around ripples in CA3. Consistent with our previous study, it also suggests that feedforward inhibition from CA3 to CA1 may play an important role in controlling the population event size in CA1 and may contribute to the post-ripple hyperpolarization (*Hulse et al., 2016*).

## Discussion

The results above show that the $V_m$ modulation of CA3 pyramidal neurons by entorhinal input and ripples is largely opposite to that of DG granule cells and CA1 pyramidal neurons. On the timescale of seconds, slow shifts in the $V_m$ of CA3 neurons are negatively correlated to the UP-DOWN transitions in the level of entorhinal input to DG. Consequently many CA3 neurons exhibit hyperpolarization in the UP state in contrast to DG and CA1 neurons, which are in sync with EC inputs. This subfield-specific modulation by UDS explains the slow trends in the membrane potential leading to and following awake ripples. On a sub-second timescale, both DG and CA1 neurons exhibit depolarization transients both ~100 ms before and immediately after the ripple onset, while CA3 neurons show a prominent hyperpolarization that starts to build before and reaches maximum after the ripple onset. The magnitude of the CA3 hyperpolarization scales with the size of the CA3 population burst pointing to feedback inhibition as its likely source.

### UDS differentially modulate activity across DG, CA3, and CA1

Neocortical dynamics during NREM (non-Rapid Eye Movement) sleep and anesthesia show intrinsic alternation between periods of elevated activity (UP states) and relative quiescence (DOWN states) (*Cowan and Wilson, 1994*; *Steriade et al., 1993a*). The entorhinal cortex is the major gateway linking the neocortex with the hippocampal formation and has been shown to exhibit UDS associated with bimodal membrane potential distributions of EC neurons (*Isomura et al., 2006*). Despite an absence of bimodality, the subthreshold activity and the spiking of hippocampal neurons is modulated by cortical UDS in a subfield-specific manner both in sleep and under anesthesia (*Hahn et al., 2007*; *Isomura et al., 2006*; *Sullivan et al., 2011*). We found that DG molecular layer currents exhibit UP-DOWN dynamics indicating that the EC undergoes UDS transitions in quiet wakefulness as well that modulate activity across hippocampal subfields.

Our results regarding UDS modulation in quiet wakefulness are largely consistent with corresponding observations in NREM sleep and under anesthesia, but there were also some notable differences that we highlight below. We observed pronounced modulation of both subthreshold activity and spiking at the DOWN→UP transition in all hippocampal subfields including a 50% increase in the baseline firing rate of CA3 neurons preceding the DOWN→UP transition, consistent with (*Isomura et al., 2006*) and in contrast to previous reports which found no modulation of CA3 unit activity in sleep (*Sullivan et al., 2011*) or weak and mixed modulation in anesthetized animals (*Hahn et al., 2007*). We also find that awake ripples occur essentially exclusively during the UP state, and not merely with an increased probability relative to the DOWN state (*Sullivan et al., 2011*). Furthermore, awake ripples occurred throughout the UP state and were not concentrated near the DOWN→UP transitions (*Battaglia et al., 2004*). Finally, in our data the vast majority of CA1 neurons were depolarized on the DOWN→UP transition and had elevated firing rates in the UP state in contrast to previous observations under anesthesia (*Hahn et al., 2007*; *Isomura et al., 2006*).

While the majority of recorded neurons were influenced by UDS, the nature of the modulation is subfield-specific. Granule cells in DG by and large follow the EC inputs and show sustained depolarization and firing rate increase throughout the UP state mirrored by relative hyperpolarization and reduced firing in the DOWN state. Pyramidal neurons in CA1 also depolarize and fire more on the DOWN→UP transition, but these responses are more transient than in DG. Consequently, the expected membrane potential across the CA1 population has a triangular wave shape as a function of UDS phase, unlike the square wave shape characteristic of the DG population. This triangular wave shape is almost symmetric with respect to DOWN→UP transition and as a result the $V_m$ conditional means in the UDS are very similar, despite the clear $V_m$ modulation by UDS phase. The CA1 population starts depolarizing before the DOWN→UP transition and the DG granule cells. This is reflected in

the non-causal transfer model impulse response of CA1 pyramidal cells, which is consistent with the presence of a feedback loop via the CA1→EC connection, but also suggests the presence of another excitatory source, such as CA3, that leads EC activity.

How can activity in CA3 lead given that CA3 is downstream of both the EC and the DG? The majority of CA3 pyramidal neurons are negatively correlated to DG molecular layer currents, which is surprising since CA3 and DG receive excitatory input from EC, while the mossy fibers (DG→CA3) form powerful excitatory 'detonator' synapses on the proximal dendrites of CA3 pyramidal neurons. Despite this anatomical organization suggesting that CA3 activity should follow that in EC and DG, CA3 pyramidal cells in fact show peak depolarization and elevated firing before the DOWN→UP transition, thus leading both CA1 and DG. Furthermore, a third of the CA3 population is more depolarized in the DOWN state, while the rest exhibit transient depolarization right before the DOWN→UP transition. This is reflected in the negative impulse response of the CA3 transfer model with sustained negative step response. The modulation of the population firing rate in CA3 by UDS is consistent with CA3 being responsible for the CA1 lead over DG activity.

What mechanisms may account for the CA3 behavior? One possibility is that DG and/or EC inputs provide powerful feedforward inhibition to CA3 pyramidal neurons. Indeed mossy fibers (DG→CA3) not only form large mossy terminals on CA3 pyramidal cells but also contact interneurons via filopodial extensions, providing an anatomical substrate for feedforward inhibition (*Acsády et al., 1998*). The balance between feedforward excitation and inhibition depends on the pattern of granule cell activity: low frequency activation of the mossy fibers results in powerful slow inhibition of CA3 pyramidal neurons while at higher frequencies an initial depolarization precedes the inhibition (*Zucca et al., 2017*). Through this mechanism the elevated DG activity in the UP state may induce a concomitant suppression in the majority of CA3 pyramidal neurons during the UP state. As granule cells reduce their firing in the DOWN state, the CA3 circuit is released from the DG-mediated feedforward inhibition and the recurrent CA3 connections may support a sustained increase in population activity. This recurrent excitation is controlled by the strong feedback inhibition present in CA3. The non-causal positive component of the CA3 transfer model impulse response and the timing of CA3 firing relative to the DOWN→UP transition indicate that CA3 may play an important role in ushering the subsequent UP state by providing excitation to EC via CA1. These observations are inconsistent with a feedforward activation of the trisynaptic pathway, suggesting a more complex interplay of intrahippocampal and perforant pathways.

## Membrane potential dynamics around ripples in quiet wakefulness

The recurrent circuit of CA3 has long been hypothesized to function as an autoassociative memory network (*Marr, 1971*) and to support the buildup of population activity underlying the ripple generation process (*Buzsáki, 2015*). Indeed, acute silencing of Schaffer collaterals during wakefulness abolishes ripples (*Davoudi and Foster, 2019*; *Yamamoto and Tonegawa, 2017*). While the membrane potential of CA1 pyramidal neurons near ripples has been shown to exhibit a gradual ramping and a transient depolarization followed by a prolonged inhibition (*Hulse et al., 2016*), the subthreshold dynamics of DG granule cells and CA3 pyramidal neurons near ripples in awake animals had not been fully characterized.

We observed that the ripple-triggered membrane potential of hippocampal neurons is modulated on a timescale of seconds, with DG granule cells showing depolarization, CA3 pyramidal neurons hyperpolarization, and CA1 cells exhibiting weaker modulation. Since ripples occur almost exclusively in the UP state and the slow $V_m$ dynamics of hippocampal cells are modulated by UDS in a subfield-specific fashion, we hypothesized and confirmed that the slow $V_m$ responses near ripples can be qualitatively accounted for by the UDS influence on hippocampal cells.

How does UDS influence hippocampal network excitability? It has been proposed that during NREM sleep hippocampal dynamics exhibit a stable, but excitable quiescent state, such that activity fluctuations can produce a transient population excitation representing a ripple (*Levenstein et al., 2019*). Our data indicate that in quiet wakefulness UDS modify hippocampal network excitability because no ripples are produced in the DOWN state. We illustrate this behavior in a mean firing rate model in the framework described in *Levenstein et al., 2019*, featuring two (EC and CA3) adapting recurrent neural populations (*Figure 8*). In the model the EC population influences CA3 activity by providing a net inhibitory input as well as by modulating the strength of the CA3 recurrent excitation (*Figure 8A*).

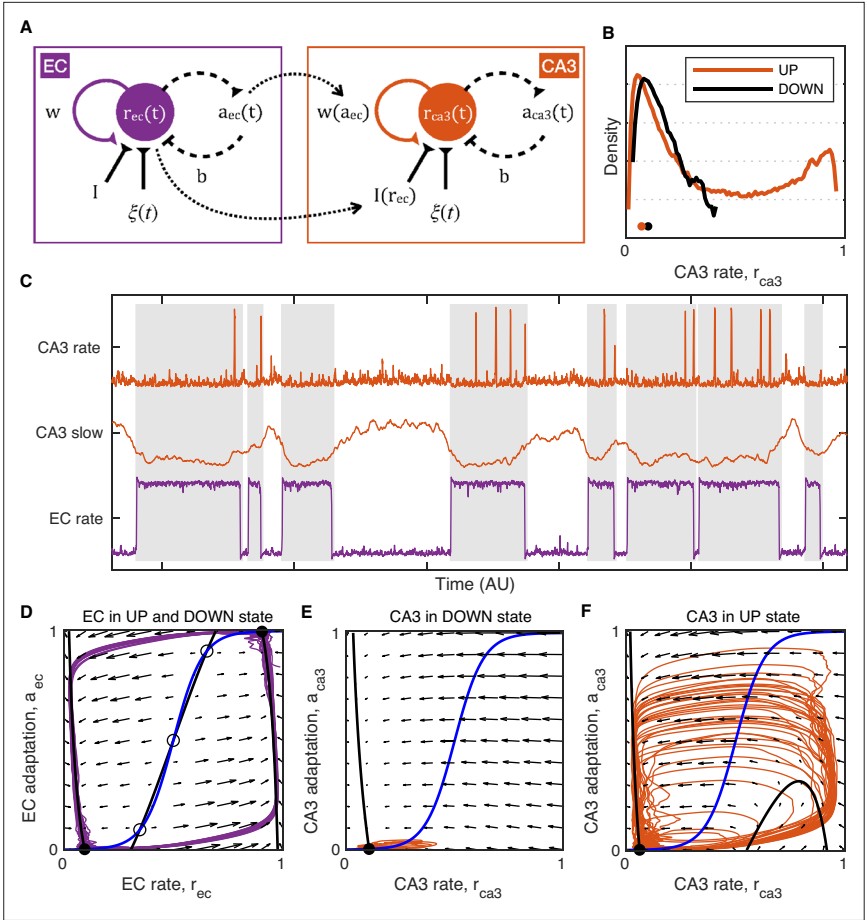

**Figure 8.** Model of UP and DOWN states (UDS) control of CA3 network excitability. (**A**) Idealized model of entorhinal cortex (EC) and CA3 adapting recurrent neural populations. EC activity provides net inhibition to the CA3 population via $I(r_{ec})$, possibly due to feedforward inhibition or indirect influence via DG, and modulates the strength of CA3 recurrent excitation via $w(a_{ac})$, modeling the effects of UDS-dependent shifts in cholinergic tone on CA3 synaptic transmission. (**B**) Probability density of the simulated CA3 population rate $r_{ca3}$ during EC UP and DOWN states. Notice that the median CA3 population rate is higher in the DOWN than the UP state (black and orange dots) despite the presence of population bursts (rate values near 1) restricted only to the UP state. (**C**) Example model simulation demonstrating that the EC population exhibits UP and DOWN dynamics while the CA3 population produces transient population bursts ('ripples') restricted to the UP state (gray segments). The slow component of the CA3 rate is magnified in the middle to show that mean CA3 activity is higher during the DOWN state and lower during the UP state when population bursts occur. (**D–F**) Phase plane plots of the model dynamics. The model evolution is governed by the velocity fields displayed as arrows. The model stable (filled circles) and unstable (open circles) fixed points occur at the intersections of the population rate (black) and adaptation (blue) nullclines. Model trajectories are plotted in purple and orange. (**D**) EC dynamics exhibit two stable fixed points (black circles) corresponding to the UP and DOWN states with noise fluctuations driving transitions between them. (**E**) CA3 dynamics in the EC DOWN state exhibit a single stable fixed point (black circle) and are not excitable, i.e., noise fluctuations cannot trigger a spike in the population rate. (**F**) CA3 dynamics in the EC UP state exhibit a single stable fixed point (black circle) at a lower population rate level than in the DOWN state (**E**), but are excitable, i.e., noise fluctuations can trigger population spikes ('ripples').

The online version of this article includes the following source data for figure 8:

**Source data 1.** Model output time series.

In the real circuit the latter influence may be due to UDS-dependent changes in neuromodulatory inputs, such as cholinergic tone, as exhibited, for example, by pedunculopontine cholinergic neurons (*Mena-Segovia et al., 2008*). The majority of the cholinergic input to the hippocampus originates in the medial septum and optogenetic stimulation of septal ChAT-positive neurons suppresses ripple generation (*Hunt et al., 2018*; *Vandecasteele et al., 2014*). Acetylcholine is known to inhibit the

efficacy of recurrent synaptic transmission in CA3 by acting on presynaptic muscarinic receptors in the associational–commissural fiber system (*Hasselmo et al., 1995*; *Hasselmo and Schnell, 1994*; *Vogt and Regehr, 2001*) and cholinergic tone is presumably at its lowest during ripple generation in the EC UP state. Thus the efficacy of CA3 recurrent connections together with dentate and entorhinal input to CA3 and the associated feedforward inhibition can act as bifurcation parameters for the CA3 network dynamics that change network excitability thereby preventing ripple occurrence during the cortical DOWN state (*Figure 8E–F*). This is counterintuitive because many CA3 neurons are more active and depolarized in the DOWN state or near the DOWN→UP transition and so, according to the stochastic-refractory model of ripple initiation (*Schlingloff et al., 2014*) the rate of ripple occurrence should coincide with the modulation of CA3 activity by UDS, which is contradicted by the fact that ripples occur in the UP state. In our model, CA3 network excitability is shown to be restricted to the EC UP state when the mean CA3 rate is lower compared to the DOWN state (*Figure 8B–C*). This is possible because of the push-pull influences of the net inhibitory input which lowers CA3 network excitability, and the CA3 recurrent strength potentiation which increases it. Hence, in the UP state the CA3 network is inhibited but excitable, while in the DOWN state it is disinhibited but not excitable (*Figure 8*).

These data suggest that increased inhibition in CA3 may be a prerequisite for ripple initiation. This is consistent with in vitro work showing that reducing $GABA_A$-mediated inhibition in hippocampal slices abolishes spontaneously occurring sharp wave-ripple events in CA3 (*Bazelot et al., 2016*; *Ellender et al., 2010*; *Schlingloff et al., 2014*). The relative hyperpolarization in CA3 during the UP state may reflect the role of certain local (*Katona et al., 2014*; *Viney et al., 2013*) or long-range projecting interneurons (*Basu et al., 2016*; *Unal et al., 2018*) in suppressing the initiation of ripples. According to one view, ripple generation may be the result of disinhibition; however, our data does not offer clear evidence for widespread disinhibition in CA3 preceding the population burst (*Evangelista et al., 2020*). Importantly, ripple generation by disinhibition does not account for the lack of ripples in the DOWN state, despite the elevated neuronal activity in CA3. Its origin notwithstanding, membrane hyperpolarization in CA3 may influence ripple initiation by affecting voltage-gated conductances and thereby changing the excitability of CA3 neurons to make them more likely to fire or burst in response to transient depolarizing input. Changes in inhibition across the CA3 network may shift the region of ripple initiation, allowing for reactivation of diverse memory traces stored within the CA3 network.

On a faster timescale, the average $V_m$ responses near ripples reveal two consistent features. First, the majority of CA3 neurons exhibit a brief (~300 ms) hyperpolarization locked to the ripple onset in addition to the broad (~3 s) UDS-mediated hyperpolarization. This brief hyperpolarization grows with the size of the CA3 population burst, quantified by the associated sharp wave amplitude, therefore pointing to feedback inhibition as its source. This suggests that feedback inhibition is a consistent feature of the buildup process in CA3. This inhibition likely arises from several classes of interneurons that have been shown to exhibit elevated firing around ripples in both CA3 and CA1 (*Klausberger et al., 2003*; *Somogyi et al., 2014*; *Tukker et al., 2013*). However, the distinct $V_m$ responses we observed around ripples indicate that inhibition is tuned differently in CA3 and CA1. In particular, in CA1 inhibition imposes oscillatory patterning on the $V_m$ that rides on a wave of depolarization, while in CA3 inhibition summates to produce a net hyperpolarization. In both areas however, fast fluctuations in the membrane potential persist throughout the ripple period allowing CA3 neurons to fire despite the net hyperpolarization. These results suggest that the relative gain of feedback inhibition is greater than that of recurrent excitation for the majority of the CA3 neurons and hence the population burst can only build up by recruiting the CA3 neurons most strongly connected to a sparse active subset. This may reflect a winner-take-all mechanism for controlling both the sparsity and the specificity of the neuronal sequences activated in CA3 during a ripple. The growing inhibition during the course of a ripple also provides a mechanism for ripple termination.

Second, DG granule cells exhibit two transient depolarizations of comparable amplitude ~100ms before and immediately after the ripple onset. Similarly timed features are present in the $V_m$ of CA1 pyramidal neurons, albeit the pre-ripple depolarization is significantly smaller than the post-ripple one. In CA1, both depolarizations are associated with sharp waves of proportional magnitudes in stratum radiatum (Figrue 4A in *Hulse et al., 2016*), implicating CA3 as the source for both. The DG activation is consistent with the presence of a backprojection from CA3 to DG (*Scharfman, 2007*; *Szabo et al., 2017*) and can influence ripple-associated CA3 activity, consistent with previous lesion

studies (*Sasaki et al., 2018*). These results indicate a long and orchestrated ripple initiation process in the awake state, extending beyond the roughly 50ms period of excitatory activity buildup that proceeds ripples in vitro (*Schlingloff et al., 2014*).

While there are consistent features within each subfield with respect to UDS and ripple modulation, there is also variability of responses across cells. Resting $V_m$ and cell location along the proximodistal and deep-superficial axes could be factors influencing the UDS and ripple modulation. Indeed, previous studies have shown a gradient of connectivity and intrinsic cell properties along the proximodistal axis in CA3 (*Sun et al., 2017*), and differential responses of CA1 neurons in deep and superficial layers of CA1 (*Valero et al., 2015*; *Valero and de la Prida, 2018*). Analysis of UP-DOWN modulation as a function of resting $V_m$ and proximodistal location of each cell (*Figure 5—figure supplement 4*) did not reveal any significant trends. Furthermore, similar analysis of ripple modulation did not reveal significant trends either, except for the fact that the most depolarized CA3 cells tend to hyperpolarize most during ripples, consistent with the fact that these cells are furthest away from the GABAa reversal potential and experience the highest driving force (*Figure 6—figure supplement 1*). However, our recordings do not span the full proximodistal axis and may hence not be ideally suited to test the dependence of our results on proximodistal location. Furthermore, we record from multiple neurons in each experiment (*Figure 1—figure supplement 1*) which does not allow us to unequivocally identify the depth of each neuron within the cell layer. Future experiments can provide more conclusive evidence concerning the factors that explain the variability of subthreshold modulation by UDS and ripples across the hippocampal subfields.

These results provide novel insights into the processes of ripple initiation, build up, and termination in awake animals. Ripples occur exclusively in UP states characterized by increased entorhinal inputs to DG and associated with pronounced hyperpolarization of CA3 pyramidal cells. This suggests that broad inhibition in CA3 may be a prerequisite for ripple initiation. Furthermore, DG and CA1 pre-ripple responses suggest that ripples are not initiated as isolated events within CA3, but are the culmination of extended interplay across multiple areas. This may reflect the role of cortical inputs in influencing the neuronal patterns replayed by the hippocampus during awake ripples, consistent with their role in spatial decision-making. Finally, growing hyperpolarization in CA3 throughout the course of a ripple suggests that feedback inhibition is a key feature of ripple buildup as well as termination. This may reflect a winner-take-all mechanism, by which a few neurons that fire overcoming a background of inhibition in UP states, further suppress all other neurons via feedback inhibition, ensuring sparseness and selectivity of transient network pattern activation.

## Materials and methods

### Head fixation surgery

The methods used were the same as those described in our previous publications (*Hulse et al., 2017*; *Hulse et al., 2016*). Briefly, male C57Bl/6 mice (Charles River Laboratories) were surgically implanted with a light-weight, stainless steel ring using dental cement. A stainless steel reference wire was implanted over the cerebellum for LFP silicon probe recordings. The locations of future craniotomies for probe and whole-cell recordings were marked. Probe recording coordinates were anterioposterior (AP)/mediolateral (ML): –1.7/1.75 in the left hemisphere for DG; AP/ML: –1.7/2.0 in the left hemisphere for CA1; and AP/ML: –2.15/0.84 in the right hemisphere for CA3. Whole-cell recording coordinates were: AP/ML: –1.7/0.65 in the left hemisphere for DG; AP/ML: –1.9/1.5 in the left hemisphere for CA1; and AP/ML: –2.15/3.1 in the right hemisphere for CA3. All coordinates are reported in mm, all AP and ML coordinates are with respect to bregma. Following surgery, mice were returned to their home cage, maintained on a 12 hr light/dark cycle, and given access to food and water ad libitum. Ibuprofen (0.2 mg/mL) was added to the water as a long-term analgesic. Mice were given at least 48 hr to recover before the day of the experiment.

### Exposure surgery

On the day of the recording, while mice (4–8 weeks old) were anesthetized with 1% isoflurane and head-fixed in the stereotaxic apparatus, two small craniotomies (~200–500 μm) were made at the previously marked locations and the dura was resected over these locations. A recording chamber was secured on top of the head-fixation device and filled with pre-oxygenated (95% $O_2$, 5% $CO_2$),

filtered (0.22 μm) artificial cerebrospinal fluid (aCSF) containing (in mM): 125 NaCl, 26.2 NaHCO3, 10 Dextrose, 2.5 KCl, 2.5 CaCl2, 1.3 MgSO4, and 1.0 NaH2PO4. The procedure was carried out ~6 hr before recordings began. To control for the effect of same day anesthesia exposure on the recordings, in two mice the dura resection procedure was carried out 3 days before the recordings and the mice were habituated to head-fixation on the spherical treadmill for 2 days.

### Awake, in vivo recordings

Awake, in vivo electrophysiological recordings were carried out following previously described methods (*Hulse et al., 2017*; *Hulse et al., 2016*). Mice were head-fixed on a spherical treadmill secured on an air table (TMC). A single-shank, 32-site silicon probe (NeuroNexus) with 100 μm site spacing was inserted in the coronal plane (~15° angle pointing toward the midline) to a depth of 2600–3400 μm and was adjusted for reliably recording LFP ripple oscillations in CA1. To find the rough target depth of whole-cell recording, at first juxtacellular recordings (*Pinault, 1996*) were performed with pipettes filled with aCSF. Whole-cell patch-clamp recordings were performed with a blind-patch approach (*Margrie et al., 2002*; *Pinault, 1996*) in current clamp mode after the target depth had been identified. Pipettes had a resistance of 5–8 MΩ and were filled with an internal solution containing (in mM): 115 K-Gluconate, 10 KCl, 10 NaCl, 10 Hepes, 0.1 EGTA, 10 Tris-phosphocreatine, 5 KOH, 13.4 Biocytin, 5 Mg-ATP, and 0.3 Tris-GTP. The internal solution had an osmolarity of 300 mOsm and a pH of 7.27 at room temperature. Pipettes are pulled from borosilicate capillaries (OD: 1.0 mm, ID: 0.58 mm; Sutter Instrument Company) using a Model P-2000 puller (Sutter Instrument Company) and inserted into the brain in the coronal plane with a~15° angle pointing away from the midline. Recordings were made using a Multiclamp 700B amplifier (Molecular Devices). The $V_m$ was not corrected for liquid junction potential. Capacitance neutralization was set prior to establishing the GΩ seal. Access resistance was estimated online by fitting the voltage response to hyperpolarizing current steps (*Hulse et al., 2016*). The input resistance was estimated by subtracting the access resistance from the ratio of the change in $V_m$ produced by the current step over the magnitude of injected current.

### Signal acquisition

All electrophysiological signal acquisition was performed with Labview (National Instruments). Electrophysiological signals were sampled simultaneously at 25 kHz with 24 bit resolution using AC (PXI-4498, internal gain: 30 dB, range: +/-316 mV) or DC-coupled (PXIe-4492, internal gain: 0 dB, range: +/-10 V) analog-to-digital data acquisition cards (National instruments) with built-in anti-aliasing filters for extracellular and intracellular/juxtacellular recordings, respectively.

### Histology and imaging

To identify the recorded neurons, histology and imaging were performed, as previously described (*Horikawa and Armstrong, 1988*; *Hulse et al., 2017*; *Hulse et al., 2016*). Following the experiment, mice were deeply anesthetized with 5% isoflurane, decapitated, and the brain extracted to 4% paraformaldehyde (PFA). Brains were fixed at 4°C in 4% paraformaldehyde overnight and transferred to 0.01 M (300 mOsm) phosphate buffered saline (PBS) the next day. Up to one week later, brains were sectioned coronally (100 μm) on a vibrating microtome (Leica), permeabilized with 1% Triton X-100 (v/v) in PBS for 1–2 hr, and incubated overnight at room temperature in PBS containing avidin-fluorescein (1:200, Vector Laboratories), 5% (v/v) normal horse serum (NHS), and 0.1% Triton X-100. Sections were rinsed in PBS between each step. The next day, sections containing biocytin stained neurons were identified on an inverted epifluorescent microscope (Olympius IX51) for further immunohistochemical processing. Sections underwent immunohistochemical staining against calbindin (CB) and parvalbumin (PV) to aid locating the recorded neurons in the hippocampus. Sections containing biocytin-stained neurons were first incubated in blocking solution containing 5% NHS, 0.25% Triton X-100, and 0.02% (wt/v) sodium azide in PBS. Next, slices were incubated in PBS containing primary antibodies against CB (Rabbit anti-Calbindin D-28k, 1:2000, Swant) and PV (Goat anti-parvalbumin, 1:2000, Swant) overnight. After thorough rinsing in PBS, slices were incubated in PBS containing secondary antibodies CF543 donkey anti-rabbit (1:500, Biotium) and CF633 donkey anti-goat (1:500, Biotium). Processed slices were rinsed and mounted in antifading mounting medium (EverBrite, Biotium). Stained slices were imaged on an inverted confocal laser-scanning microscope (LSM 710 & LSM 880, Zeiss).

## Data analysis

### $V_m$ decomposition

Spikes were detected by identifying local maxima in the broadband membrane potential with peak prominence of at least 15 mV and width of at most 10 ms. Cells with spike thresholds exceeding –37 mV were considered to be possibly damaged and were therefore excluded from subsequent analyses. The subthreshold membrane potential ($V_m$) was computed by interpolating the membrane potential over periods with an action potential starting from 3ms before to 5ms after the spike peak. The subthreshold $V_m$ signal was then low-pass filtered (Parks-McClellan optimal equiripple FIR filter, 250–350 Hz transition band) and downsampled to 2083 Hz. The $V_m$ signal was decomposed into fast, slow, and drift components as follows. First, the slow $V_m$ component ($V_{m,slow}$) was obtained by median filtering $V_m$ with a 1 s window. Next, the fast $V_m$ component ($V_{m,fast}$) was obtained as the residual $V_m$ after subtraction of $V_{m,slow}$. A drift component ($V_{m,drift}$) was obtained by smoothing $V_{m,slow}$ with a 60 s boxcar kernel. Finally, $V_{m,slow}$ was detrended by subtracting $V_{m,drift}$, which included the resting membrane potential as well as any $V_m$ changes on the timescale of minutes (*Figure 2—figure supplement 3*). By construction $V_m = V_{m,fast} + V_{m,slow} + V_{m,drift}$ and the components contain different frequency bands of the subthreshold membrane potential.

### Ripple detection

Ripples were detected as transient increases in power in the ripple frequency band of the LFP from the probe site located in the CA1 pyramidal cell layer. Ripple power was estimated by band-pass filtering the LFP trace (80–250 Hz), smoothing its square with a Gaussian kernel (10 ms), and taking the square root. Candidate ripple events were identified as segments for which the ripple power was more than 3 s.d. above the mean. Segments that were less than 55ms apart were merged, and after the merging step segments of length less than 20 ms were rejected as artifacts. A reference recording site away from the CA1 cell layer that does not exhibit ripples was identified and the same procedure was applied for ripple detection on this reference channel. Events detected in both the CA1 and the reference LFP trace were rejected as artifacts.

### Current source density (CSD) estimation

Local field potentials (LFPs) were recorded from a 32-site silicon probe with 100 µm site spacing positioned so that sites spanned all of neocortex, area CA1, the DG, and parts of the thalamus. LFPs were bandpass filtered between 1 Hz and 1 KHz and downsampled to 2083 Hz. Channels with recording artifacts were excluded from the CSD analysis. Laminar current source densities were estimated with ~17 µm resolution using a robust version of the one-dimensional inverse CSD spline method (*Pettersen et al., 2006*). In particular, the forward matrix (relating CSDs to LFPs) was computed as usual, but the inverse matrix (relating LFPs to CSDs) was computed using ridge regression with a regularization parameter set by a cross-validation procedure. The spatial smoothness of the CSD estimate was automatically controlled by the regularization parameter and therefore no further spatial smoothing was applied. The anatomical laminae in CA1 and DG were determined using a combination of histological reconstruction of the probe track, electrophysiological markers (ripples, sharp waves, and dentate spikes), and the CSD covariance structure. DG CSD activity was computed by first averaging the rectified CSD signals from the vertical extent (~200 µm) of the suprapyramidal molecular layer of DG (giving the 'rectified DG CSD'), then smoothing with a 1 s median filter, and finally converting to a z-score. By construction DG CSD activity only reflects the rate and magnitude of transient synaptic activity and not any slow-varying (<1 Hz) transmembrane currents which are inherently difficult to capture with AC-coupled recordings (*Brankack et al., 1993*). Using DG CSD activity, we identify UP states as time periods when the rate and amplitude of EC input current transients, rather than the DC level, increases, in accordance with previous publications (*Isomura et al., 2006*). We further validated that the extracted UP/DOWN states reflect modulation of pupil radius and ripple rate, quantities that are independently measured.

### Transfer model estimation

Linear transfer models were estimated after taking the z-scored DG CSD activity as model input and the slow $V_m$ component of the subthreshold membrane potential of a given cell as output. The FIR was estimated from the input-output data using a regularized nonparametric procedure (impulseest

in Matlab System Identification Toolbox with tuned and correlated 'TC' kernel used for regularization) (*Chen et al., 2012*). The input-output data were first downsampled to ~20 Hz and the order of the FIR was set to 75, corresponding to 3.6 s duration. The procedure automatically estimated FIR values at negative delays (up to –0.86 s) and non-zero filter values at negative delays indicated that the slow $V_m$ component led DG CSD activity. Once the impulse responses were estimated the corresponding step responses were simulated by feeding the model with a step input. Low order autoregressive with extra input (ARX) models were also estimated in a similar manner and led to qualitatively similar results as the FIR models (data not shown).

## UP and DOWN state segmentation

UDS were identified from the z-scored DG molecular layer CSD activity using a HMM (*McFarland et al., 2011*). First, a binary Gaussian mixture was fit to the distribution of DG CSD values (*Figure 2C*). Next, the mixture components were used to initialize the emission probability distributions of a two state (UP and DOWN) HMM and then the state transition and emission probabilities were estimated from the DG CSD time series data. Finally, the most likely sequence of states given the observed DG CSD time series, downsampled to 4 Hz, were computed with the Viterbi algorithm. The resulting Viterbi path was used to assign a UDS phase to each time point based on its position relative to the nearest UDS transition times. The method was unsupervised and did not require user tuning of any model parameters.

## Significance testing of UDS modulation

To test whether UP-DOWN state had an effect on subthreshold membrane potential we sampled the $V_m$ signal every 2 s (in order to remove any serial correlation between retained samples) and grouped the retained samples according to UP-DOWN state. We then compared the means of the UP and DOWN groups using a parametric two-sample t-test with unpooled variance and also compared the medians using the non-parametric Wilcoxon rank sum test, which yielded consistent results. To test whether the variance of the fast $V_m$ component in the UDS was different we used a similar procedure, but the signal was sampled every 1 s (since the fast component decorrelated faster) and the variances of the UP and DOWN groups were compared using a parametric two-sample F test, as well as a non-parametric two-sample Ansari-Bradley test, which yielded consistent results. To test whether UP-DOWN state had an effect on the firing rate of a given cell we compared the observed spike counts in the UDS against the expected counts given the observed mean firing rate using Pearson's chi-square test of goodness of fit. Finally, to test whether UDS phase had an effect on $V_m$ we grouped the sampled signal in 20 bins according to UDS phase and checked if the mean of each group was different from 0 using a one-sample Z-test with significance level reduced by a factor of 20 to account for the multiple comparisons. Modulation by UDS phase was deemed significant if 5 or more of the 20 groups had means significantly different than 0.

## Adapting recurrent neural population model

When uncoupled, the EC and CA3 population dynamics follow the Wilson-Cowan type r-a model studied in detail in *Levenstein et al., 2019*. Briefly, the mean firing rate of the EC population ($r_{ec}$) evolves under activity-driven adaptation ($a_{ec}$) according to the equations:

$$\tau_r \dot{r} = -r + R_\infty \left( wr - ba + I + \xi\left(t\right) \right) \tag{1}$$

$$\tau_a \dot{a} = -a + A_\infty\left(r\right) \tag{2}$$

with sigmoid activation functions $R_\infty\left(x\right)$ and $A_\infty\left(x\right)$ given by the logistic curve,

$$f\left(x\right) = \frac{1}{1+e^{-k\left(x-x_0\right)}} \tag{3}$$

where $k=1, x_o = 5$ for $R_\infty\left(x\right)$ and $k=15, x_o = 0.5$ for $A_\infty\left(x\right)$. The time constants $\tau_r = 1$ and $\tau_a = 25$ are dimensionless and set the arbitrary units (AU) of the time axis. For the EC population the strength of recurrent excitation ($w$), adaptation weight ($b$), and the tonic drive ($I$) are all constant parameters with the following values: $w = 6.8, b = 1, I = 2.1$. The model is excited by stochastic fluctuations $\xi\left(t\right)$ given by an Ornstein–Uhlenbeck process

$$d\xi_t = -\theta\xi_t dt + \sigma\sqrt{2\theta}dW_t \tag{4}$$

with parameters $\theta = 0.05$ and $\sigma = 0.1$, corresponding to a time constant of 20 and steady-state standard deviation of $\sigma$. The mean firing rate of the CA3 population evolves under the same *Equations (1-4)*, but the drive is now a linear function of EC activity

$$I(t) = I_0 - I_g r_{ec}(t) \tag{5}$$

with $I_0 = 2.5$ and $I_g = 0.6$, so that elevated EC activity generates a net inhibitory drive to CA3. The strength of recurrent excitation in CA3 is also a linear function of the EC adaptation parameter

$$w(t) = w_0 + w_g a_{ec}(t) \tag{6}$$

with $w_0 = 3.5$ and $w_g = 2.5$, so that the EC UP state leads to an increase in the strength of CA3 recurrent excitation. The stochastic fluctuations exciting the CA3 model have $\sigma = 0.25$ and are unrelated to those in EC.

### Pupil analysis

Pupil videos were analyzed using DeepLabCut (*Mathis et al., 2018*). A subset of frames were manually labeled with 8 points around the pupil and these points were subsequently extracted with DeepLabCut for the remaining frames. A core set of points with likelihood >0.7 was detected for each frame and a circle was fit using the Taubin SVD-based method for frames with more than 3 core points (*Taubin, 1991*). Blinks were detected as abrupt drops in likelihood across the detected points. Datasets with mean likelihood across detected points above 0.95 and with more than 97% valid frames were selected for analysis. Proper extraction of pupil diameter and blinks was confirmed by visual inspection.

## Acknowledgements

We thank Lee-Peng Mok for help with histological processing and immunohistochemistry, Maria Papadopoulou and Stijn Cassenear for help with imaging, insightful discussions and feedback on the manuscript, and Kevin Shan for help with the data processing pipeline and insightful discussion. Confocal imaging was performed at the Caltech Biological Imaging Facility. This work was supported by a Vannevar Bush Faculty Fellowship, the Mathers Foundation, the McKnight Foundation, and NIH grant RO1MH113016.

## Additional information

### Funding

| Funder | Grant reference number | Author |
| --- | --- | --- |
| DoD | Vannevar Bush Faculty Fellowship | Athanassios G Siapas |
| NIH | RO1MH113016 | Athanassios G Siapas |
| Mathers Foundation | | Athanassios G Siapas |
| McKnight Foundation | | Athanassios G Siapas |

The funders had no role in study design, data collection and interpretation, or the decision to submit the work for publication.

### Author contributions

Koichiro Kajikawa, Brad K Hulse, Conceptualization, Data curation, Formal analysis, Investigation, Methodology, Software, Validation, Visualization, Writing – review and editing; Athanassios G Siapas, Evgueniy V Lubenov, Conceptualization, Data curation, Formal analysis, Funding acquisition, Investigation, Methodology, Project administration, Resources, Software, Supervision, Validation, Visualization, Writing – original draft, Writing – review and editing

**Author ORCIDs**
Koichiro Kajikawa http://orcid.org/0000-0002-8626-4407
Athanassios G Siapas http://orcid.org/0000-0001-8837-678X
Evgueniy V Lubenov http://orcid.org/0000-0002-1099-944X

**Ethics**
All procedures were approved by the Institutional Animal Care and Use Committee (IACUC) at Caltech (protocols 1465, 1771) and conformed to National Institutes of Health guidelines.

**Decision letter and Author response**
Decision letter https://doi.org/10.7554/eLife.69596.sa1
Author response https://doi.org/10.7554/eLife.69596.sa2

## Additional files

**Supplementary files**
• Transparent reporting form

**Data availability**
All data analyzed during this study are included in the manuscript and supporting files.

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
