## [Editor Report]

This paper combines intracellular and extracellular recordings in the hippocampus in awake mice to investigate the initiation of sharp wave-ripples, synchronous bursts of activity thought to support memory replay. They report a specific hyperpolarization of the pyramidal cells in the CA3 subfield while the dentate granule cells or CA1 pyramidal cells are depolarized. This paper will be of interest to system neuroscientists interested in the cellular and network mechanisms of memory formation.

---

## [Decision Letter]

**Decision letter after peer review:**

Thank you for submitting your article "Inhibition is the hallmark of CA3 intracellular dynamics around awake ripples" for consideration by *eLife*. Your article has been reviewed by 3 peer reviewers, and the evaluation has been overseen by a Reviewing Editor and Laura Colgin as the Senior Editor. The following individuals involved in review of your submission have agreed to reveal their identity: Liset M de la Prida (Reviewer #1); Jerome Epzstein (Reviewer #3).

Essential revisions:

The present study addresses the question of sharp wave-ripples initiation using patch-clamp recordings of principal cells in different hippocampal subfields (CA3, CA1 and the dentate gyrus – DG) combined with extracellular recordings in awake head-fixed mice as well as computational modeling. CA3 pyramidal cells are depolarized and spike at the DOWN/UP transition (with some cells depolarizing even earlier) and then progressively hyperpolarize during the course of the UP state while DG cells and CA1 pyramidal cells tend to depolarize and fire throughout the UP state. Hence, the study suggests that switches between a dominant inhibitory excitable state and a disinhibited non-excitable state control the intra-hippocampal dynamics during UP/DOWN transitions. While the three reviewers are quite enthusiastic about the study, they have all raised a number of concerns regarding the presentation of the data and of the manuscript. The most critical points are summarized below:

1) More information regarding the actual recordings should be reported, including the mean resting membrane potential (true voltage, not z-scored), input resistance of the neurons, burst propensity, morphological validation of cell types, and number of neurons per area. This will help the reader to judge the quality of the dataset and to compare with previously published studies in the field.

2) The true membrane potential should be reported not only for quality criteria, but also as an explanatory factor for 'inhibitory/excitatory' responses due to different driving forces.

3) The recordings were done the same day as anaesthesia, leading to potential confounds when comparing with anaesthesia-free preparation. This should be clearly stated. Furthermore, there are discrepancies with previously reported findings using other techniques, especially regarding the firing of CA3 and CA1 neurons during ripples and the duration of UP and DOWN states. The study should discuss clearly why, and whether anaesthesia and/or the use of whole cell recording technique may explain these differences.

4) The precise location of each neuron within each hippocampal subfield is also not reported. This is quite important as this may explain part of the variability observed near UP/DOWN transitions and around ripples.

5) It is unclear why the ripple onset is used and not the more commonly used ripple peak. This will help compare the results with previously published papers in the field.

6) The CSD analysis should be improved: in its present form, an absolute phase cannot be ascertained and the relationship with entorhinal/cortical inputs and ongoing animal's behavior should be clarified. Moreover, the average CSD around times of DOWN/UP transition and ripples should be reported. It is also unclear whether the coherence of the CSD in the DG and CA3 membrane potential is specific to slow frequencies or was observed across the frequency spectrum. Finally, baseline of CSD estimated from AC-coupled LFP can be misleading. Some of these issues can be addressed by using a blind source separation instead.

7) The data regarding pupil diameter and eye blink should more thoroughly analyzed. Importantly, it should be clarified whether they were collected across animals or from a single experiment. The unit of the coherence with DG CSD amplitude is unclear (Figure 2, S1B).

8) Can the dynamics of ripple trains (which usually occur in "bursts") explain the increase in DG cells up to 150ms before ripple onset?

9) The title of the manuscript is misleading. The present data don't support that CA3 inhibition is the only hallmark of ripple generation and, in general, the study focuses more on the relationship of member potentials with UP/DOWN state fluctuation than with ripples.

10) The manuscript lacks some other important details, e.g. how cells were ranked in Figures3-6 and the number of observations in each figure legend. Please see below comments from the reviewers for more detailed remarks.

*Reviewer #1 (Recommendations for the authors):*

This is a very interesting paper describing membrane potential dynamics of hippocampal principal cells during UP/DOWN transitions and sharp-wave ripples. Using whole-cell in combination with linear LFP recordings in head-fixed awake mice, the authors show striking differences of membrane potential responses in principal cells from the dentage gyrus, CA3 and CA1 sectors. They propose that switches between a dominant inhibitory excitable state and a disinhibited non-excitable state control the intra-hippocampal dynamics during UP/DOWN transitions. While data show clear trends and some of the conclusions are supported by the analysis, the authors may need to clarify some potential confounding which can actually impact interpretation.

1 – All the analysis is based in z-scored membrane potential responses but the mean resting membrane potential is never reported. For DG granule cells recorded in awake conditions, the membrane potential is usually hyperpolarized so that most of the effect may be due to reversed GABAA mediated currents. Similarly, for those cells exhibiting the non-expected polarization during UP/DOWN states there may be drifts around reversal potentials explaining their behavior. Moreover, regional trends on passive and active membrane parameters and connectivity can actually explain part of the variability. A longitudinal comparison of state Vm and spikes in Figure 5 suggests that some of the largest depolarized responses are not correlated with firing. Authors should evaluate this angle, ideally showing the distribution of membrane potential values across cells and regions and confronting this with the different membrane potential responses.

2 – While there are some trends for each hippocampal regions, there is also individual variability across cells during UP/DOWN transitions (Figure 5) and near ripples (Figure 6). What part of this variability can be explained by proximodistal and/or deep-superficial differences of cell location and identity? Can authors provide some morphological validation, even if in only a subset of cells? For CA3, proximodistal heterogeneity for intrinsic properties and entorhinal input responses are well documented in intracellular recordings both in vitro and in vivo. What is the location of CA3 cell contributing to this study? For CA1 cells, deep-superficial trends of GABAergic perisomatic inhibition and connectivity with input pathways dominate firing responses. Regarding DG cells, are all they from the upper blade?

3- AC-coupled LFP recordings cannot provide unambiguous identification of the sign of phasic CSD signals, because fluctuations accompanying UP/DOWN states alter the baseline reference. This is actually the case, given changes of membrane potential accompanying UP/DOWN transitions. I recommend reading Brankack et al., 1993 doi: 10.1016/0006-8993(93)90043-m. The authors should acknowledge this limitation and discuss how it could influence their results. One potential solution to get rid of this effect is using principal/independent component analysis for blind source separation.

Other important considerations:

4 – Following the previous comment, are all LFP signals recorded along CA1-DG? Since CSD currents should be estimated along the somatodendritic axis of cells receiving laminar input arrangements, it is possible that poor correlation with CA3 cell signals may reflect poor alignment with the corresponding input sources. Authors should try evaluating or discussing this confound.

5 – Figure 2 and the supplementary Figure 1 provide very interesting data integrating behavioral variables such as pupil diameter and eye blinks. It is unclear however whether this comes from a single experiment. It is also unclear how the pupil and eye blink dynamics was measured and evaluated. These figures establish the methods used to evaluate global variables, such as DG CSD signals, which are then correlated with single cell membrane potentials across figures, but no quantification is provided. If this is the result of a single experiment, then authors should consider replication in at least a subset of cases.

6 – There are some misconceptions regarding connectivity of different cell types. The claim that different fields receive similar inputs from entorhinal cortex may not be correct. The entorhinal inputs to DG and CA3a,b come from LII stellate cells, while excitatory inputs to CA1 are from layer II pyramidal cells (direct inputs) and layer III pyramidal cells (indirect through feedforward inhibition).

7 – One potentially controversial issue emerging from data is how could CA3 cells initiate sharp-wave ripples given they are mostly inhibited. Figure 8 provides a useful framework for discussion based on how dynamical fluctuations of modulatory influences and entorhinal inputs determine different regimes of excitability. The authors may want to consider two additional angles: that the sharp-wave ripples represent actually micro states emerging randomly from different generators. This could fit well with the idea that they reactivate several memory traces, and therefore there should not be a dominant initiator. In this context, inhibition plays major role in firing selectivity across the entire network. A second point is that state-dependent modulation in excitability could actually shift the ripple generator within a given region. While stochastic excitability fluctuations are more likely to trigger population bursts from the recurrently interconnected CA3 regions, the degree of connectivity follows a proximodistal distribution. Actually, circuit computation of proximal CA3c cells is more likely engaged in pattern separation whereas distal CA3a cells play more a pattern completion role. Similarly CA2 cells recurrently connected with CA3a cells may also trigger CA1 events. Hence, there is not such a marked difference between regions, but a biological continuum of intrinsic excitability, connectivity and computation.

Miscellaneous comments

8 – The title brings the focus to the intracellular dynamics around ripples while most of results are actually related to UP/DOWN states.

9 – Figure 2-sup.Figure 1B, is coherence expressed in absolute values? Since pupil diameter (magenta) and DG CSD magnitude are somehow anti-correlated one expect values from -1 to 1. Please, clarify.

10 – Figure 3 and 4, how are cell ranked in these two figures versus Figure 5 and 6? To ease interpretation it may be useful to use consistent ranking criteria across figures. Please, specify in captions. A side note here: cell variability within CA3 and CA1 as shown in Figure 3A3 and Fig6 may be reminiscent of proximodistal and/or deep-superficial trends reported previously. While authors may lack the ability too properly evaluate these effects (see point 2 above), it should be considered and discussed.

11 – In all figures reporting single cell data, please consider providing the N in captions.

*Reviewer #2 (Recommendations for the authors):*

In order to fully evaluate the findings the data need to be presented in a more clear and accountable way and the analyses need to be extended.

The ripple onset is not a reliable enough marker, I would want to see all plots remade with the peak time for comparison.

The authors should show all the filled cells, intrinsic physiology, or something for each and every cell to make sure they are all the same type, healthy, etc, before claiming as such. The CA3 spiking plots around SWRs look like two types of cells. Is that the two principal types as described by Hunt et al? Or is one group interneurons, or sick principal cells?

Most of the paper is about UDS not SWRs: consider changing the title or focus.

Why aren't mice trained to be head fixed before the day of recording, as in other studies?

Why were the craniotomies not made at least 24 hours before recording so that you know there is no residual anesthesia?

*Reviewer #3 (Recommendations for the authors):*

1) Membrane potential values should be indicated for each intracellular recording in Fig. 1 and average Vm in UP and DOWN state should be reported for all recorded cell types (DGC, CA1 and CA3).

2) P8 “the behavior of the example cell in Figure 2B is surprising because CA3 receive nearly identical excitatory inputs as DG” I would downstate this statement because CA3 and DGCs are very different cell types notably DGC are electrically much more compact and inputs from EC contact them closer to the cell body and AIS. Furthermore, CA3 will receive concomitant inputs from DGCs during EC UP state which contact them through mossy fibers that have an overall inhibitory impact on CA3 pyramidal cell at low frequency because mossy fiber boutons strongly recruit feedforward inhibition through fillopoda extensions (Acsady et al., J Neurosci1998; Henze et al., Nat Neurosci 2002; Mori et al., Nature 2004).

3) The fact that Isomura et al., 2006 already reported a depolarization of CA3 pyramidal cells during EC DOWN state should be clearly acknowledged. From their paper: “In contrast to neo/paleocortical neurons, the maximum relative depolarizationof most CA3 cells coincided with the neocortical DOWN state.” and their Fig. 5A.

4) Could you report the number of recorded cells in each subfield in the main text and their passive/active electrical properties (firing threshold, input resistance, baseline Vm etc..)? Generally I think more numbers and P values should be reported throughout the manuscript.

5) P8 “ the majority of DG and CA1 cells” could you give actual numbers and proportion ? How was the positive/negative response of a neuron determined ? Did you used a specific threshold? Some correlations in CA1 (Fig 3. C3) occur with a large delay (2s) and are close to zero but yet are depicted as blue (negatively modulated) dots. How did you assess their significance? I think it is fair to acknowledge that CA1 pyramidal cells have a mixed coherence with DG CSD between that of DGCs and CA3 pyramidal cells.

6) Fig. 7A3 if average traces corresponding to small and big ripples are of different colors this should be specified in the legend. The black curves prevent seeing the curve associated to big ripple (I think) and statistical difference could be highlighted by horizontal bars at the top of the graph.

7) In the discussion “We observed pronounced modulation and spiking at the DOWN-UP transition in contrast to previous study” but Isomura et al., 2006 report an increase firing of CA3 pyramidal cells around the DOWN-UP transition under anesthesia (Fig. 5 D, E).

8) “DG activity is more similar to that in CA1 than in CA3” but the Vm profile with a sharp depolarization at the transition between DOWN-UP and then progressive hyperpolarization throughout the UP state is similar in CA3 and CA1 (Fig. 5D1).

---

## [Author Response]

Essential revisions:The present study addresses the question of sharp wave-ripples initiation using patch-clamp recordings of principal cells in different hippocampal subfields (CA3, CA1 and the dentate gyrus – DG) combined with extracellular recordings in awake head-fixed mice as well as computational modeling. CA3 pyramidal cells are depolarized and spike at the DOWN/UP transition (with some cells depolarizing even earlier) and then progressively hyperpolarize during the course of the UP state while DG cells and CA1 pyramidal cells tend to depolarize and fire throughout the UP state. Hence, the study suggests that switches between a dominant inhibitory excitable state and a disinhibited non-excitable state control the intra-hippocampal dynamics during UP/DOWN transitions. While the three reviewers are quite enthusiastic about the study, they have all raised a number of concerns regarding the presentation of the data and of the manuscript. The most critical points are summarized below:1) More information regarding the actual recordings should be reported, including the mean resting membrane potential (true voltage, not z-scored), input resistance of the neurons, burst propensity, morphological validation of cell types, and number of neurons per area. This will help the reader to judge the quality of the dataset and to compare with previously published studies in the field.

We added Figure 1 —figure supplement 4, which now describes the mean resting membrane potential, input resistance, burst propensity, and spikes per burst for the recorded cells. These data are provided in Figure 1 – source data 1 together with a recording identifier that can be used to link each cell to all other figure panels and data files. We further added Figure 1—figure supplement 1, which provides examples of morphological information for our recordings, Figure 1 —figure supplement 2 that shows examples of bursts from morphologically identified neurons, and Figure 1 —figure supplement 3 that shows the locations of recorded cells.

The number of cells was (indirectly) reported as the number of rows in Figures 3-7. The numbers of cells included in the revised version of the manuscript are now stated explicitly: 22 DG cells, 32 CA3 cells, and 32 CA1 cells.

2) The true membrane potential should be reported not only for quality criteria, but also as an explanatory factor for 'inhibitory/excitatory' responses due to different driving forces.

We added Figure 5 —figure supplement 4 that includes the resting V_m_ and proximodistal locations of all cells in relation to their UP-DOWN modulation. We did not detect any significant trends with respect to brain state modulation. Furthermore, we added Figure 6 —figure supplement 1 that includes the resting V_m_ and proximodistal location of all cells in relation to their ripple modulation. This figure shows that the most depolarized cells tend to hyperpolarize most during ripples, consistent with the fact that these cells are furthest away from the GABAA reversal potential and experience the highest driving force. This trend was only statistically significant for CA3 neurons and no other significant trends were detected.

3) The recordings were done the same day as anaesthesia, leading to potential confounds when comparing with anaesthesia-free preparation. This should be clearly stated. Furthermore, there are discrepancies with previously reported findings using other techniques, especially regarding the firing of CA3 and CA1 neurons during ripples and the duration of UP and DOWN states. The study should discuss clearly why, and whether anaesthesia and/or the use of whole cell recording technique may explain these differences.

The main surgery for implanting the head-fixation apparatus and marking the coordinates for multisite and pipette insertion was carried out at least two days before the experiment. On the day of the experiment animals were briefly lightly anesthetized (<1 hr, at <1% isoflurane at 1 lit/min) for the sole purpose of resecting the dura at the two sites for multisite probe and pipette insertion. This procedure was carried out on the same day as the experiment in order to minimize the time the brain was exposed and optimize the quality of the recordings. Experiments began at least six hours after this short procedure. Furthermore, animals were given time to get familiarized with the behavioral apparatus before recordings began and showed no signs of distress.

Previous studies show that about 95% of isoflurane is eliminated within minutes by exhalation (Holaday et al., 1975). The further elimination of isoflurane proceeds with a fast phase with half-time of about 7-9 min and a slower phase with half-time of about 100-115 min (Chen et al., 1992), with the faster phase reflecting elimination from the brain (Litt et al., 1991). Given these considerations there should be negligible residual isoflurane from the short anesthesia six hours later when recordings are initiated.

In order to further investigate whether the short and light anesthesia during the day of recordings has any effect on the results reported in the paper, we carried out additional experiments in which we performed the surgery, including dura removal, 3 days before the recording session. The animals were habituated under head-fixation on the spherical treadmill for two hour periods each of the two days following the surgery. On the third day after surgery, we carried out recordings without any surgical procedures or anesthesia. The durations of UP and DOWN states without same day anesthesia were similar to those obtained in our previous experiments (Figure 2—figure supplement 4). The additional CA3 whole-cell recordings obtained in these new experiments have the same hyperpolarization features typical of our previous recordings. These additional experiments argue that the brief anesthesia on the day of recordings has no significant effect on the results.

The reviewers also refer to potential discrepancies in the firing of CA3 and CA1 cells during ripples and the duration of UP and DOWN states with respect to previous studies. The CA1 neurons in this study depolarize and elevate their firing around ripples, consistent with previous intracellular and extracellular recordings. Our study reveals hyperpolarization of the majority of CA3 cells while only a small fraction is depolarized around ripples. This is consistent with the sparse activation of CA3 around ripples previously reported with extracellular studies. The overall firing rate change of CA3 neurons around ripples is a balance between the firing rate elevation of the small subset of activated cells and the net decrease in firing across the rest of the population. Since the baseline firing rate of CA3 pyramidal neurons in quiet wakefulness and sleep is low, the ripple-associated inhibition may not be readily observable in the spiking of individual CA3 neurons due to a “floor effect”. The overall rate of CA3 neurons we record increases before ripple onset, consistent with previous extracellular studies (Figure 6D4). The subthreshold hyperpolarization of the majority of CA3 neurons provides novel insights into the mechanisms ensuring sparse and selective activation of the CA3 population around ripples.

During sleep and under anesthesia the transitions between UP and DOWN states are largely driven by internal brain dynamics as the role of external stimuli is minimized. The dwell times can be particularly consistent under anesthesia, although the exact values depend on the nature of the anesthetic agent and the depth of anesthesia (Torao-Angosto et al., 2021). In contrast, in quiet wakefulness the duration of UP and DOWN states will presumably be influenced also by the behavior of the animal, its attentional state, and external stimuli and therefore need not be the same as under anesthesia or sleep. Here we simply report the presence of UP-DOWN dynamics during quiet wakefulness. Whether or not the mechanisms responsible are the same or different than those operating under sleep and anesthesia remains to be seen, as the latter mechanisms themselves have not been fully identified. To provide validation that the extracted UP and DOWN states in quiet wakefulness indeed correspond to genuine brain states, we show that the pupil diameter and ripple rates, which are independently measured, are strongly modulated around the identified UP and DOWN states.

4) The precise location of each neuron within each hippocampal subfield is also not reported. This is quite important as this may explain part of the variability observed near UP/DOWN transitions and around ripples.

We added figure 1 —figure supplement 3 that shows the locations of recorded neurons.

As described in point (2) above, we added Figure 5 —figure supplement 4 that includes the resting V_m_ and proximodistal locations of all cells in relation to their UP-DOWN modulation. We did not detect any significant trends with respect to brain state modulation. Furthermore, we added Figure 6 —figure supplement 1 that includes the resting V_m_ and proximodistal locations of all cells in relation to their ripple modulation. Again, we did not detect significant trends with regards to proximodistal location, although we would like to note that our recordings do not span the full proximodistal axis and may hence not be ideally suited to test the dependence of our results on proximodistal location.

5) It is unclear why the ripple onset is used and not the more commonly used ripple peak. This will help compare the results with previously published papers in the field.

We added a Figure 6 —figure supplement 2, which shows that the modulation around peak ripple power is the same as the modulation around ripple start, except for a small time shift due to the fact that ripple power peaks shortly after ripple start. Our focus on ripple onset facilitates characterizing the timing of pre-ripple activity, such as the V_m_ depolarization observed before ripple onset for DG and CA1 neurons.

6) The CSD analysis should be improved: in its present form, an absolute phase cannot be ascertained and the relationship with entorhinal/cortical inputs and ongoing animal's behavior should be clarified. Moreover, the average CSD around times of DOWN/UP transition and ripples should be reported. It is also unclear whether the coherence of the CSD in the DG and CA3 membrane potential is specific to slow frequencies or was observed across the frequency spectrum. Finally, baseline of CSD estimated from AC-coupled LFP can be misleading. Some of these issues can be addressed by using a blind source separation instead.

The reviewers are correct in pointing out that the DC and very low frequency components of the CSD are not readily measured experimentally with silicon probes. The DC half-cell potential of metal microelectrodes is dominated by the properties of the electrode/electrolyte interface and its magnitude (in the hundreds of millivolts) dwarfs normal brain signals. As the electrode-electrolyte interface is not stable, neither is the near-DC electrode potential and hence it cannot be easily filtered out or rejected even in DC-coupled recordings, although this approach has been successful for studying high-amplitude (tens of millivolts) LFP shifts associated with spreading depression (Nasretdinov et al., 2017).

With this consideration in mind, our understanding of the reviewers’ point is summarized in Author response image 1. As the figure illustrates, although the low frequency component of the CSD is not directly measured with our AC-coupled recordings, the modulation of the magnitude of high-frequency transients can provide an envelope signal with arbitrarily low frequency content. Experimentally, we observe that the rate and magnitude of high frequency transients of the DG CSD is modulated by a low frequency (< 1 Hz) “envelope” signal, which itself exhibits alternation between low and high levels, in accordance with previous publications (Isomura et al., 2006). Since the high frequency transients originate from postsynaptic activity due to EC input, high levels of the envelope signal indicate elevated EC activity (UP state), while low levels indicate reduced EC activity (DOWN state). In the original submission we referred to this envelope signal as “DG CSD magnitude”, which may have been confusing. In the revised manuscript we use “DG CSD activity” instead to remove any suggestion that the low frequency CSD was directly measured. We further validated that the extracted UP/DOWN states reflect modulation of pupil radius and ripple rate, quantities that are independently measured.

**Author response image 1. sa2fig1:** Deriving slow envelope signal from AC coupled recordings. (A) In this example the true CSD signal contains both a slow component (8 Hz) and a fast component (80 Hz) that is amplitude modulated by the slow component. Such phase-amplitude coupling is well known between theta and gamma oscillations in the hippocampus. The true CSD shows a current sink with time-varying magnitude. (B) The power spectral density (PSD) estimate of the signal in (A) shows both the slow (8 Hz) and fast (three peaks near 80 Hz) components. (C) Assume LFP recordings are obtained with a high-pass filter that has eliminated the slow component. Consequently, the estimated CSD signal contains only fast fluctuations. Furthermore, instead of a time-varying current sink it shows quickly alternating sinks and sources (both negative and positive values). The slow component can be visualized as the amplitude envelope (interrupted red line) of the signal. (D) PSD estimate shows that the slow component is absent from the extracted CSD signal. (E) Rectifying the CSD estimate (black) and then filtering (red) approximately recovers the true slow component (red interrupted). This is how the DG CSD activity signal is obtained. (F) PSD estimate of the rectified and filtered CSD signal recovers the slow component (interrupted red vertical line).

We included Figure 2 —figure supplement 2, which shows that the coherence of the rectified DG CSD and V_m_ is high preferentially for low frequencies and this profile is consistent across cells and subfields.

We added Figure 2 —figure supplement 6, which shows the average CSD triggered on ripples.

Finally, we added Figure 2 —figure supplement 7, which shows the rectified DG CSD triggered on UP/DOWN transition times and ripples.

7) The data regarding pupil diameter and eye blink should more thoroughly analyzed. Importantly, it should be clarified whether they were collected across animals or from a single experiment. The unit of the coherence with DG CSD amplitude is unclear (Figure 2, S1B).

We added Figure 2 —figure supplement 3 showing pupil diameter around UP and DOWN transitions across datasets. Furthermore, we added Figure 2 —figure supplement 2 showing the rate of ripple and blink occurrence as a function of UDS phase across datasets. The relationships shown in Figure 2, and previous S1B,D are consistent across the datasets.

We also extended the analysis of the coherence of V_m_ with the rectified DG CSD across all datasets (Figure 2 —figure supplement 1). We report the magnitude-squared coherence, whose values range from 0 to 1. We now clarify this in the caption of Figure 2 —figure supplement 1.

8) Can the dynamics of ripple trains (which usually occur in "bursts") explain the increase in DG cells up to 150ms before ripple onset?

We added Figure 7 —figure supplement 2 that compares V_m_ aligned to the onset of isolated single ripples vs. ripple doublets. The pre-ripple depolarization in DG and CA1 is similar for isolated ripples and ripple doublets arguing against the hypothesis that pre-ripple responses are a reflection of ripple bursts.

9) The title of the manuscript is misleading. The present data don't support that CA3 inhibition is the only hallmark of ripple generation and, in general, the study focuses more on the relationship of member potentials with UP/DOWN state fluctuation than with ripples.

We changed the title to: *“Up-Down states and ripples differentially modulate membrane potential dynamics across DG, CA3, and CA1 in awake mice”.*

10) The manuscript lacks some other important details, e.g. how cells were ranked in Figures3-6 and the number of observations in each figure legend. Please see below comments from the reviewers for more detailed remarks.

We added all missing information in the figures and corresponding captions.

Reviewer #1 (Recommendations for the authors):This is a very interesting paper describing membrane potential dynamics of hippocampal principal cells during UP/DOWN transitions and sharp-wave ripples. Using whole-cell in combination with linear LFP recordings in head-fixed awake mice, the authors show striking differences of membrane potential responses in principal cells from the dentage gyrus, CA3 and CA1 sectors. They propose that switches between a dominant inhibitory excitable state and a disinhibited non-excitable state control the intra-hippocampal dynamics during UP/DOWN transitions. While data show clear trends and some of the conclusions are supported by the analysis, the authors may need to clarify some potential confounding which can actually impact interpretation.1 – All the analysis is based in z-scored membrane potential responses but the mean resting membrane potential is never reported. For DG granule cells recorded in awake conditions, the membrane potential is usually hyperpolarized so that most of the effect may be due to reversed GABAA mediated currents. Similarly, for those cells exhibiting the non-expected polarization during UP/DOWN states there may be drifts around reversal potentials explaining their behavior. Moreover, regional trends on passive and active membrane parameters and connectivity can actually explain part of the variability. A longitudinal comparison of state Vm and spikes in Figure 5 suggests that some of the largest depolarized responses are not correlated with firing. Authors should evaluate this angle, ideally showing the distribution of membrane potential values across cells and regions and confronting this with the different membrane potential responses.

We added Figure 1 —figure supplement 4, which now describes the mean resting membrane potential, input resistance, burst propensity, and spikes per burst for the recorded cells. These data are provided in Figure 1 – source data 1 together with a recording identifier that can be used to link each cell to all other figure panels and data files. We further added Figure 1 —figure supplement 1, which provides examples of morphological information for our recordings, Figure 1 —figure supplement 2 that shows examples of bursts from morphologically identified neurons, and Figure 1 —figure supplement 3 that shows the locations of recorded cells.

In addition, we added Figure 5 —figure supplement 4 that includes the resting V_m_ and proximodistal location of cells in relation to their UP-DOWN modulation. We did not detect any significant trends with respect to brain state modulation. DG cells are more hyperpolarized compared to CA3 and CA1 cells and are closest to the reversal potential for GABAA (Figure 1 —figure supplement 4). The lack of any clear trends with respect to the resting V_m_ suggests that drifts around the GABAA reversal potential are unlikely to be a major factor driving variability in the observed UDS modulation.

2 – While there are some trends for each hippocampal regions, there is also individual variability across cells during UP/DOWN transitions (Figure 5) and near ripples (Figure 6). What part of this variability can be explained by proximodistal and/or deep-superficial differences of cell location and identity? Can authors provide some morphological validation, even if in only a subset of cells? For CA3, proximodistal heterogeneity for intrinsic properties and entorhinal input responses are well documented in intracellular recordings both in vitro and in vivo. What is the location of CA3 cell contributing to this study? For CA1 cells, deep-superficial trends of GABAergic perisomatic inhibition and connectivity with input pathways dominate firing responses. Regarding DG cells, are all they from the upper blade?

We now provide morphological validation for a subset of cells (Figure 1 —figure supplement 1). Since we patch multiple cells in each experiment it is not possible to unequivocally determine their depth within the cell layer, although it is possible to confirm that they are granule cells or pyramidal cells in experiments where all labeled cells are principal neurons (Figure 1 —figure supplement 1). In addition, we added Figure 1 —figure supplement 3 that shows the proximodistal locations of recorded cells. With respect to the DG cells 20/22 are from the upper blade, with only two granule cells recorded in the lower blade (Figure 1 —figure supplement 3).

We added Figure 5 —figure supplement 4 that includes the resting V_m_ and proximodistal location of each cell as a function of UP-DOWN modulation. We did not detect any significant trends with respect to UDS modulation.

In addition, we added Figure 6 —figure supplement 1 that includes the resting V_m_ and proximodistal location of each cell as a function of ripple modulation. This figure shows that the most depolarized CA3 cells tend to hyperpolarize most during ripples, consistent with the fact that these cells are furthest away from the GABAA reversal potential and experience the highest driving force. No other significant trends were detected, although we would like to note that our recordings do not span the full proximodistal axis and may hence not be ideally suited to test the dependence of our results on proximodistal location.

3- AC-coupled LFP recordings cannot provide unambiguous identification of the sign of phasic CSD signals, because fluctuations accompanying UP/DOWN states alter the baseline reference. This is actually the case, given changes of membrane potential accompanying UP/DOWN transitions. I recommend reading Brankack et al., 1993 doi: 10.1016/0006-8993(93)90043-m. The authors should acknowledge this limitation and discuss how it could influence their results. One potential solution to get rid of this effect is using principal/independent component analysis for blind source separation.

We acknowledge the inherent limitations of AC-coupled recordings in regards to CSD analysis (Brankack et al., 1993). However, we do not believe these limitations affect our analysis or results for the reasons illustrated in Author response image 1. Specifically, we do not attempt to measure the low frequency (< 1 Hz) CSD content directly. Instead, we extract the envelope of the rectified fast CSD transients. In the original submission we referred to this envelope signal as “DG CSD magnitude”, which may have been confusing. In the revised manuscript we use “DG CSD activity” instead to remove any suggestion that the low frequency CSD signal was directly measured. Notice that because of the rectification step the envelope signal is insensitive to the actual polarity of the fast transient CSD fluctuations. Using the envelope, we identify UP states as time periods when the rate and amplitude of EC input current transients, rather than the DC level, increases, in accordance with previous publications (Isomura et al., 2006). We further validated that the extracted UP/DOWN states reflect modulation of pupil diameter and ripple rate, quantities that are independently measured.

Other important considerations:4 – Following the previous comment, are all LFP signals recorded along CA1-DG? Since CSD currents should be estimated along the somatodendritic axis of cells receiving laminar input arrangements, it is possible that poor correlation with CA3 cell signals may reflect poor alignment with the corresponding input sources. Authors should try evaluating or discussing this confound.

All LFP signals are recorded along CA1-DG in the same way across all experiments: the location of the multisite extracellular probe is fixed across experiments, while the patch pipette target changes for experiments targeting different subfields. Hence, the resulting CSD signals are similar across experiments providing a common reference against which membrane potential changes across the subfields are evaluated. The correlations between CA3 V_m_ and CSD signals in DG are just as strong as the correlations between CA1 and DG V_m_ and CSD signals, but of opposite polarity (Figure 3).

5 – Figure 2 and the supplementary Figure 1 provide very interesting data integrating behavioral variables such as pupil diameter and eye blinks. It is unclear however whether this comes from a single experiment. It is also unclear how the pupil and eye blink dynamics was measured and evaluated. These figures establish the methods used to evaluate global variables, such as DG CSD signals, which are then correlated with single cell membrane potentials across figures, but no quantification is provided. If this is the result of a single experiment, then authors should consider replication in at least a subset of cases.

We extended the analysis of the coherence of V_m_ with DG CSD across the datasets (new Figure 2 —figure supplement 1). Furthermore, we added Figure 2 —figure supplement 2, which includes analysis of the rate of occurrence of ripples and blinks as a function of UP and DOWN states across datasets. Finally, we added Figure 2 —figure supplement 3 with analysis of pupil diameter across datasets and a methods section on how pupil diameter was measured. The relationships shown in Figure 2, S1B,D of the previous submission are consistent across the datasets.

6 – There are some misconceptions regarding connectivity of different cell types. The claim that different fields receive similar inputs from entorhinal cortex may not be correct. The entorhinal inputs to DG and CA3a,b come from LII stellate cells, while excitatory inputs to CA1 are from layer II pyramidal cells (direct inputs) and layer III pyramidal cells (indirect through feedforward inhibition).

The reviewer correctly points out that different sets of EC cells project to DG, CA3, CA1 which need not have similar firing properties. Given these considerations we removed characterization of EC inputs to DG, CA3, and CA1 as “similar” in the text, as this was not essential for any of the scientific arguments of the manuscript.

7 – One potentially controversial issue emerging from data is how could CA3 cells initiate sharp-wave ripples given they are mostly inhibited. Figure 8 provides a useful framework for discussion based on how dynamical fluctuations of modulatory influences and entorhinal inputs determine different regimes of excitability. The authors may want to consider two additional angles: that the sharp-wave ripples represent actually micro states emerging randomly from different generators. This could fit well with the idea that they reactivate several memory traces, and therefore there should not be a dominant initiator. In this context, inhibition plays major role in firing selectivity across the entire network. A second point is that state-dependent modulation in excitability could actually shift the ripple generator within a given region. While stochastic excitability fluctuations are more likely to trigger population bursts from the recurrently interconnected CA3 regions, the degree of connectivity follows a proximodistal distribution. Actually, circuit computation of proximal CA3c cells is more likely engaged in pattern separation whereas distal CA3a cells play more a pattern completion role. Similarly CA2 cells recurrently connected with CA3a cells may also trigger CA1 events. Hence, there is not such a marked difference between regions, but a biological continuum of intrinsic excitability, connectivity and computation.

We have revised the discussion to further describe how inhibition may play a role in enhancing firing selectivity and the reactivation of distinct memory traces. We have also added discussion of the potential effects of the proximodistal location, depth, and baseline membrane potential in the diversity of UDS and ripple modulation.

Miscellaneous comments8 – The title brings the focus to the intracellular dynamics around ripples while most of results are actually related to UP/DOWN states.

We changed the title to: “Up-Down states and ripples differentially modulate membrane potential dynamics across DG, CA3, and CA1 in awake mice”.

9 – Figure 2-sup.Figure 1B, is coherence expressed in absolute values? Since pupil diameter (magenta) and DG CSD magnitude are somehow anti-correlated one expect values from -1 to 1. Please, clarify.

We report the magnitude-squared coherence, whose values range from 0 to 1, as we now specify in the caption of Figure 2 —figure supplement 1.

10 – Figure 3 and 4, how are cell ranked in these two figures versus Figure 5 and 6? To ease interpretation it may be useful to use consistent ranking criteria across figures. Please, specify in captions. A side note here: cell variability within CA3 and CA1 as shown in Figure 3A3 and Fig6 may be reminiscent of proximodistal and/or deep-superficial trends reported previously. While authors may lack the ability too properly evaluate these effects (see point 2 above), it should be considered and discussed.

In Figures 3, 4, 6, and 7 cells from each area are ordered the same way: by the first principal component coefficient of the ripple-triggered average (RTA) response matrices displayed in Figure 6A3,B3,C3 (RTA rank) In Figure 5 cells are ordered by the first principal component coefficient of the UP-DOWN state (UDS) mean V_m_ matrices (UDS rank) displayed in Figure 5A1,B1,C1. We have now updated the captions to provide this important information.

11 – In all figures reporting single cell data, please consider providing the N in captions.

The number of cells was (indirectly) reported as the number of rows in Figures 3-7. We now report the number of cells explicitly: 22 DG cells, 32 CA3 cells, and 32 CA1 cells.

Reviewer #2 (Recommendations for the authors):In order to fully evaluate the findings the data need to be presented in a more clear and accountable way and the analyses need to be extended.The ripple onset is not a reliable enough marker, I would want to see all plots remade with the peak time for comparison.

We added a Figure 6 —figure supplement 2, which shows that the modulation around peak ripple power is the same as the modulation around ripple start, except for a small time shift due to the fact that the ripple power peaks shortly after ripple start. Our focus on ripple onset facilitates characterizing the timing of pre-ripple activity, such as the V_m_ depolarization observed before ripple onset for DG and CA1 neurons.

The authors should show all the filled cells, intrinsic physiology, or something for each and every cell to make sure they are all the same type, healthy, etc, before claiming as such. The CA3 spiking plots around SWRs look like two types of cells. Is that the two principal types as described by Hunt et al? Or is one group interneurons, or sick principal cells?

We added Figure 1 —figure supplement 3, which now describes the mean resting membrane potential, input resistance, burst propensity, and spikes per burst for the recorded cells. We further added Figure 1 —figure supplement 1, which provides examples of morphological information for our recordings, and Figure 1 —figure supplement 2, which shows examples of bursts from morphologically identified neurons. All the cells we patched in CA3 exhibit complex bursts with comparable levels of bursting propensity (Figure 1 —figure supplements 4). The properties of morphologically identified cells in Figure 1 —figure supplement 1 are typical of all recorded cells (morphologically identified neurons from Figure 1 —figure supplement 1 are shown as diamonds in Figure 1—figure supplement 4, while the rest are shown as dots). There were no significant differences between the two groups (p > 0.05 t-test; p > 0.05 Wilcoxon rank sum test).

To avoid including cells with potentially abnormal activity we excluded all cells with spike threshold above -37 mV from the study (3 CA1 and 4 CA3 cells from the original submission that appeared as outliers in terms of spike threshold). Furthermore, 4 CA3 cells recorded without same day anesthesia were added to the study.

We also analyze the properties and UDS modulation of the CA3 neurons that are depolarized around ripples (Figure 6 —figure supplement 3). These neurons have comparable resting V_m_, spike thresholds, and burst propensity as the rest of the CA3 population (p > 0.05, t-test). These CA3 cells had lower firing probability in the DOWN state. The locations of the depolarized cells are distributed across CA3c,b and are not clustered compared to the rest of the cells (Author response image 2) .

**Author response image 2. sa2fig2:** Proximodistal locations of CA3 cells that depolarize during ripples. Same as Figure 1 - figure supplement 3, but CA3 cells showing depolarization in their ripple-triggered average (RTA) response are marked with black dots. There was no significant difference in the proximodistal locations of these cells compared to the rest of the CA3 population (p > 0.05, t-test).

Finally, the population of athorny cells described in Hunt et al., represents a small percentage of CA3 cells (10-20%) that are concentrated in the CA3a region, which we do not sample in our recordings. Hence, the depolarized cells are unlikely to correspond to the athorny cells reported in Hunt et al.

Most of the paper is about UDS not SWRs: consider changing the title or focus.

We changed the title to “Up-Down states and ripples differentially modulate membrane potential dynamics across DG, CA3, and CA1 in awake mice” to reflect the analysis of both UP-DOWN state transitions and ripples. The two analyses are linked as the brain state modulation accounts for the slow V_m_ modulation around ripples.

Why aren't mice trained to be head fixed before the day of recording, as in other studies?Why were the craniotomies not made at least 24 hours before recording so that you know there is no residual anesthesia?

The main surgery for implanting the head-fixation apparatus and marking the coordinates for multisite and pipette insertion was carried out at least two days before the experiment. On the day of the experiment animals were briefly lightly anesthetized (<1 hr, at <1% isoflurane at 1 lit/min) for the sole purpose of resecting the dura at the two sites for multisite probe and pipette insertion. This procedure was carried out on the same day as the experiment in order to minimize the time the brain was exposed and optimize the quality of the recordings. Experiments began at least six hours after this short procedure. Furthermore, animals were given time to get familiarized with the behavioral apparatus before recordings began and showed no signs of distress.

Previous studies show that about 95% of isoflurane is eliminated within minutes by exhalation (Holaday et al., 1975). The further elimination of isoflurane proceeds with a fast phase with half-time of about 7-9 min and a slower phase with half-time of about 100-115 min (Chen et al., 1992), with the faster phase reflecting elimination from the brain (Litt et al., 1991). Given these considerations there should be negligible residual isoflurane from the short anesthesia six hours later when recordings are initiated.

In order to further investigate whether the short and light anesthesia during the day of recordings has any effect on the results reported in the paper, we carried out additional experiments in which we performed the surgery, including dura removal, 3 days before the recording session. The animals were habituated under head-fixation on the spherical treadmill for two hour periods each of the two days following the surgery. On the third day after surgery, we carried out recordings without any surgical procedures or anesthesia. The durations of UP and DOWN states without same day anesthesia were similar to those obtained in our previous experiments (Figure 2—figure supplement 4). The additional CA3 whole-cell recordings obtained in these new experiments have the same hyperpolarization features typical of our previous recordings. These additional experiments argue that the brief anesthesia on the day of recordings has no significant effect on the results.

Reviewer #3 (Recommendations for the authors):1) Membrane potential values should be indicated for each intracellular recording in Fig. 1 and average Vm in UP and DOWN state should be reported for all recorded cell types (DGC, CA1 and CA3).

We now indicate the membrane potential values in Figure 1. Furthermore, we added Figure 1 - figure supplement 4 which now describes the mean resting membrane potential, input resistance, burst propensity, and spikes per burst for the recorded cells. We further added Figure 5 - supplement 5, which compares the average V_m_ in the UP and DOWN states for all recorded cells. Finally, we added Figure 1 - source data 1, which includes the average V_m_ in UP and DOWN states for all cells.

2) P8 “the behavior of the example cell in Figure 2B is surprising because CA3 receive nearly identical excitatory inputs as DG” I would downstate this statement because CA3 and DGCs are very different cell types notably DGC are electrically much more compact and inputs from EC contact them closer to the cell body and AIS. Furthermore, CA3 will receive concomitant inputs from DGCs during EC UP state which contact them through mossy fibers that have an overall inhibitory impact on CA3 pyramidal cell at low frequency because mossy fiber boutons strongly recruit feedforward inhibition through fillopoda extensions (Acsady et al., J Neurosci1998; Henze et al., Nat Neurosci 2002; Mori et al., Nature 2004).

We removed characterizations of EC inputs to DG and CA3 as “similar” from the text, as this was not essential for any of the results. We agree that feedforward inhibition from DG is important for understanding the observed effects. We discuss the impact of feedforward inhibition from DG in the paragraph starting with “What mechanisms may account for the CA3 behavior?” in the discussion section (last paragraph before the “Membrane Potential Dynamics Around Ripples in Quiet Wakefulness” section).

3) The fact that Isomura et al., 2006 already reported a depolarization of CA3 pyramidal cells during EC DOWN state should be clearly acknowledged. From their paper: “In contrast to neo/paleocortical neurons, the maximum relative depolarizationof most CA3 cells coincided with the neocortical DOWN state.” and their Fig. 5A.

We revised the discussion to read: “We observed pronounced modulation of both subthreshold activity and spiking at the DOWN→UP transition in all hippocampal subfields including a 50% increase in the baseline firing rate of CA3 neurons preceding the DOWN→UP transition, consistent with (Isomura et al., 2006), but in contrast to previous reports which found no modulation of CA3 unit activity in sleep (Sullivan et al., 2011) or weak and mixed modulation in anesthetized animals (Hahn et al., 2007).”

4) Could you report the number of recorded cells in each subfield in the main text and their passive/active electrical properties (firing threshold, input resistance, baseline Vm etc..)? Generally I think more numbers and P values should be reported throughout the manuscript.

The number of cells was (indirectly) reported as the number of rows in Figs. 3-7. We now report the number of cells explicitly: 22 DG cells, 32 CA3 cells, and 32 CA1 cells.

We added Figure 1 - figure supplement 4 which now describes the mean resting membrane potential, input resistance, burst propensity, and spikes per burst for the recorded cells (see also Figure 1 - source data 1). These data are provided in Figure 1 - source data 1 together with a recording identifier that can be used to link each cell to all other figure panels and data files. We further added Figure 1 - figure supplement 1 which provides examples of morphological information for our recordings, and Figure 1 - figure supplement 2 which shows examples of bursts from morphologically identified neurons. The location of recorded neurons is now reported in Figure 1 - figure supplement 3.

5) P8 “ the majority of DG and CA1 cells” could you give actual numbers and proportion ? How was the positive/negative response of a neuron determined ? Did you used a specific threshold? Some correlations in CA1 (Fig 3. C3) occur with a large delay (2s) and are close to zero but yet are depicted as blue (negatively modulated) dots. How did you assess their significance? I think it is fair to acknowledge that CA1 pyramidal cells have a mixed coherence with DG CSD between that of DGCs and CA3 pyramidal cells.

We now provide quantitative characterization of the responses of the cells in Figure 5 - supplementary figure 5 and added a methods section on “Significance testing of UDS modulation”. We now include actual numbers and proportions in the main text next to the qualitative descriptions. Finally, we included a clarification in the caption of Figure 3 that “a subset of CA1 cells exhibit negative correlations at positive lags”.

6) Fig. 7A3 if average traces corresponding to small and big ripples are of different colors this should be specified in the legend. The black curves prevent seeing the curve associated to big ripple (I think) and statistical difference could be highlighted by horizontal bars at the top of the graph.

The responses to big ripples are shown with darker colors, but are hard to distinguish because the responses to big and small ripples overlap for many time offsets. We have clarified that the differences in the responses to big ripples compared to small are shown in black. The statistically different parts are marked in Figure 7 D3.

7) In the discussion “We observed pronounced modulation and spiking at the DOWN-UP transition in contrast to previous study” but Isomura et al., 2006 report an increase firing of CA3 pyramidal cells around the DOWN-UP transition under anesthesia (Fig. 5 D, E).

We revised this sentence to read: “We observed pronounced modulation of both subthreshold activity and spiking at the DOWN→UP transition in all hippocampal subfields including a 50% increase in the baseline firing rate of CA3 neurons preceding the DOWN→UP transition, consistent with (Isomura et al., 2006), but in contrast to previous reports which found no modulation of CA3 unit activity in sleep/waking immobility (Sullivan et al., 2011) or weak and mixed modulation in anesthetized animals (Hahn et al., 2007).”

8) “DG activity is more similar to that in CA1 than in CA3” but the Vm profile with a sharp depolarization at the transition between DOWN-UP and then progressive hyperpolarization throughout the UP state is similar in CA3 and CA1 (Fig. 5D1).

Most CA1 cells are more depolarized in UP states, similar to DG cells. The reviewer is correct that CA1 cells do share a transient increase close to the DOWN to UP transition with CA3 cells. We removed this statement to avoid confusion.